# Minimax Lower Bounds for Transfer Learning with Linear and One-hidden Layer Neural Networks

**Seyed Mohammadreza Mousavi Kalan, Zalan Fabian, Salman Avestimehr,**
**and Mahdi Soltanolkotabi**
Ming Hsieh Department of Electrical Engineering
University of Southern California
California, Los Angeles 90089
`mmousavi@usc.edu,zfabian@usc.edu,avestimehr@ee.usc.edu,soltanol@usc.edu`

## Abstract

Transfer learning has emerged as a powerful technique for improving the performance of machine learning models on new domains where labeled training data may be scarce. In this approach a model trained for a *source* task, where plenty of labeled training data is available, is used as a starting point for training a model on a related *target* task with only few labeled training data. Despite recent empirical success of transfer learning approaches, the benefits and fundamental limits of transfer learning are poorly understood. In this paper we develop a statistical minimax framework to characterize the fundamental limits of transfer learning in the context of regression with linear and one-hidden layer neural network models. Specifically, we derive a lower-bound for the target generalization error achievable by any algorithm as a function of the number of labeled source and target data as well as appropriate notions of similarity between the source and target tasks. Our lowerbound provides new insights into the benefits and limitations of transfer learning. We further corroborate our theoretical finding with various experiments.

## 1   Introduction

Deep learning approaches have recently enjoyed wide empirical success in many applications spanning natural language processing to object recognition. A major challenge with deep learning techniques however is that training accurate models typically requires lots of labeled data. While for many of the aforementioned tasks labeled data can be collected by using crowd-sourcing, in many other settings such data collection procedures are expensive, time consuming, or impossible due to the sensitive nature of the data. Furthermore, deep learning techniques often are brittle and do not adapt well to changes in the data or the environment. Transfer learning approaches have emerged as a way to mitigate these issues. Roughly speaking, the goal of transfer learning is to borrow knowledge from a *source* domain, where lots of training data is available, to improve the learning process in a related but different *target* domain. Despite recent empirical success the benefits as well as fundamental limitations of transfer learning remains unclear with many open challenges:

*What is the best possible accuracy that can be obtained via any transfer learning algorithm? How does this accuracy depend on how similar the source and target domain tasks are? What is a good way to measure similarity/distance between two source and target domains? How does the transfer learning accuracy scale with the number of source and target data? How do the answers to the above questions change for different learning models?*

At the heart of answering these questions is the ability to predict the best possible accuracy achievable by any algorithm and characterize how this accuracy scales with how related the source and target data are as well as the number of labeled data in the source and target domains. In this paper we take

a step towards this goal by developing statistical minimax lower bounds for transfer learning focusing on regression problems with linear and one-hidden layer neural network models. Specifically, we derive a minimax lower bound for the generalization error in the target task as a function of the number of labeled training data from source and target tasks. Our lower bound also explicitly captures the impact of the noise in the labels as well as an appropriate notion of *transfer distance* between source and target tasks on the target generalization error. Our analysis reveals that in the regime where the transfer distance between the source and target tasks is large (i.e. the source and target are dissimilar) the best achievable accuracy mainly depends on the number of labeled training data available from the target domain and there is a limited benefit to having access to more training data from the source domain. However, when the transfer distance between the source and target domains are small (i.e. the source and target are similar) both source and target play an important role in improving the target training accuracy. Furthermore, we provide various experiments on real data sets as well as synthetic simulations to empirically investigate the effect of the parameters appearing in our lower bound on the target generalization error.

**Related work.** There is a vast theoretical literature on the problem of domain adaptation which is closely related to transfer learning (1; 2; 3; 4; 5; 6; 7). The key difference is that in domain adaptation there is no labeled target data while in transfer learning a few labeled target data is available in addition to source data. Most of the existing results in the domain adaptation literature give an upper bound for the target generalization error. For instance, the papers (8; 9) provide an upper bound on the target generalization error in classification problems in terms of quantities such as source generalization error, the optimal joint error of source and target as well as VC-dimension of the hypothesis class. A more recent work (10) generalizes these results to a broad family of loss functions using Rademacher complexity measures. Related, (11) derives a similar upper bound for target generalization error as in (8) but in terms of other quantities. Finally, the recent paper (12) generalizes the results of (8; 10) to multiclass classification using margin loss.

More closely related to this paper, there are a few interesting results that provide lower bounds for the target generalization error. For instance, focusing on domain adaptation the paper (13) provides necessary conditions for successful target learning under a variety of assumptions such as a covariate shift, similarity of unlabeled distributions, and existence of a joint optimal hypothesis. More recently, the paper (14) defines a new discrepancy measure between source and target domains, called *transfer exponent*, and proves a minimax lower bound on the target generalization error under a relaxed covariate-shift assumption and a Bernstein class condition. (15) provides a minimax lower bound for a related multi-task learning setting in sparse linear regression. (11) derives an information theoretic lower bound on the joint optimal error of source and target domains defined in (8). Most of the above results are based on a covariate shift assumption which requires the conditional distributions of the source and target tasks to be equal and the source and target tasks to have the same best classifier. In this paper, however, we consider a more general case in which source and target tasks are allowed to have different optimal classifiers. Furthermore, these results do not specifically study a neural network model. To the extent of our knowledge this is the first paper to develop minimax lower bounds for transfer learning with neural networks.

## 2 Problem Setup

We now formalize the transfer learning problem considered in this paper. We begin by describing the linear and one-hidden layer neural network transfer learning regression models that we study. We then discuss the minimax approach to deriving transfer learning lower bounds.

### 2.1 Transfer Learning Models

We consider a transfer learning problem in which there are labeled training data from a source and a target task and the goal is to find a model that has good performance in the target task. Specifically, we assume we have $n_S$ labeled training data from the source domain generated according to a source domain distribution $(\boldsymbol{x}_S, \boldsymbol{y}_S) \sim \mathbb{P}$ with $\boldsymbol{x}_S \in \mathbb{R}^d$ representing the input/feature and $\boldsymbol{y}_S \in \mathbb{R}^k$ the corresponding output/label. Similarly, we assume we have $n_T$ training data from the target domain generated according to $(\boldsymbol{x}_T, \boldsymbol{y}_T) \sim \mathbb{Q}$ with $\boldsymbol{x}_T \in \mathbb{R}^d$ and $\boldsymbol{y}_T \in \mathbb{R}^k$. Furthermore, we assume that the features are distributed as $\boldsymbol{x}_S \sim \mathcal{N}(0, \boldsymbol{\Sigma_S})$, $\boldsymbol{x}_T \sim \mathcal{N}(0, \boldsymbol{\Sigma_T})$ with $\boldsymbol{\Sigma}_S$ and $\boldsymbol{\Sigma}_T \in \mathbb{R}^{d \times d}$ denoting the covariance matrices. We also assume that the labels $\boldsymbol{y}_S / \boldsymbol{y}_T$ are generated from ground truth mappings relating the features to the labels as follows

$$\boldsymbol{y}_S = f(\boldsymbol{\theta}_S; \boldsymbol{x}_S) + \boldsymbol{w}_S \quad \text{and} \quad \boldsymbol{y}_T = f(\boldsymbol{\theta}_T; \boldsymbol{x}_T) + \boldsymbol{w}_T \qquad (2.1)$$

where $\boldsymbol{\theta}_S$ and $\boldsymbol{\theta}_T$ are the parameters of the function $f$ and $\boldsymbol{w}_S, \boldsymbol{w}_T \sim \mathcal{N}(0, \sigma^2 \mathbf{I}_k)$ represents source/target label noise. In this paper we focus on the following linear and one-hidden layer neural network models.

**Linear model.** In this case, we assume that $f(\boldsymbol{\theta}_S; \boldsymbol{x}_S) := f(\boldsymbol{W}_S; \boldsymbol{x}_S) = \boldsymbol{W}_S \boldsymbol{x}_S$ and $f(\boldsymbol{\theta}_T; \boldsymbol{x}_T) := f(\boldsymbol{W}_T; \boldsymbol{x}_T) = \boldsymbol{W}_T \boldsymbol{x}_T$ where $\boldsymbol{W}_S, \boldsymbol{W}_T \in \mathbb{R}^{k \times d}$ are two unknown matrices denoting the source/target parameters. The goal is to use the source and target training data to find a parameter matrix $\widehat{\boldsymbol{W}}_T$ with estimated label $\widehat{\boldsymbol{y}}_T = \widehat{\boldsymbol{W}}_T \boldsymbol{x}_T$ that achieves the smallest risk/generalization error $\mathbb{E}[\|\boldsymbol{y}_T - \widehat{\boldsymbol{y}}_T\|_{\ell_2}^2]$.

**One-hidden layer neural network models.** We consider two different neural network models where in one the hidden-to-output layer is fixed and in the other the input-to-hidden layer is fixed. Specifically, in the first model, we assume that $f(\boldsymbol{\theta}_S; \boldsymbol{x}_S) := f(\boldsymbol{W}_S; \boldsymbol{x}_S) = \boldsymbol{V}\varphi(\boldsymbol{W}_S \boldsymbol{x}_S)$ and $f(\boldsymbol{\theta}_T; \boldsymbol{x}_T) := f(\boldsymbol{W}_T; \boldsymbol{x}_T) = \boldsymbol{V}\varphi(\boldsymbol{W}_T \boldsymbol{x}_T)$ where $\boldsymbol{W}_S, \boldsymbol{W}_T \in \mathbb{R}^{\ell \times d}$ are two unknown weight matrices, $\boldsymbol{V} \in \mathbb{R}^{k \times \ell}$ is a fixed and known matrix, and $\varphi$ is the ReLU activation function. Similarly in the second model, we assume that $f(\boldsymbol{\theta}_S; \boldsymbol{x}_S) := f(\boldsymbol{V}_S; \boldsymbol{x}_S) = \boldsymbol{V}_S \varphi(\boldsymbol{W} \boldsymbol{x}_S)$ and $f(\boldsymbol{\theta}_T; \boldsymbol{x}_T) := f(\boldsymbol{V}_T; \boldsymbol{x}_T) = \boldsymbol{V}_T \varphi(\boldsymbol{W} \boldsymbol{x}_T)$ with $\boldsymbol{V}_S, \boldsymbol{V}_T \in \mathbb{R}^{k \times \ell}$ two unknown weight matrices and $\boldsymbol{W} \in \mathbb{R}^{\ell \times d}$ a known matrix. In both cases the goal is to use the source and target training data to find the unknown target parameter weights ($\widehat{\boldsymbol{W}}_T$ or $\widehat{\boldsymbol{V}}_T$) that achieve the smallest risk/generalization error $\mathbb{E}[\|\boldsymbol{y}_T - \widehat{\boldsymbol{y}}_T\|_{\ell_2}^2]$. Here, $\widehat{\boldsymbol{y}}_T = \boldsymbol{V}\varphi(\widehat{\boldsymbol{W}}_T \boldsymbol{x}_T)$ in the first model and $\widehat{\boldsymbol{y}}_T = \widehat{\boldsymbol{V}}_T \varphi(\boldsymbol{W} \boldsymbol{x}_T)$ in the second.

## 2.2 Minimax Framework for Transfer Learning

We now describe our minimax framework for developing lower bounds for transfer learning. As with most lower bounds, in a minimax framework we need to define a class of transfer learning problems for which the lower bound is derived. Therefore, we define $(\mathbb{P}_{\boldsymbol{\theta}_S}, \mathbb{Q}_{\boldsymbol{\theta}_T})$ as a pair of joint distributions of features and labels over a source and a target task, that is, $(\boldsymbol{x}_S, \boldsymbol{y}_S) \sim \mathbb{P}_{\boldsymbol{\theta}_S}$ and $(\boldsymbol{x}_T, \boldsymbol{y}_T) \sim \mathbb{Q}_{\boldsymbol{\theta}_T}$ with the labels obeying (2.1). In this notation, each pair of a source and target task is parametrized by $\boldsymbol{\theta}_S$ and $\boldsymbol{\theta}_T$. We stress that over the different pairs of source and target tasks, $\boldsymbol{\Sigma}_S, \boldsymbol{\Sigma}_T$, and $\sigma^2$ are fixed and only the parameters $\boldsymbol{\theta}_S$ and $\boldsymbol{\theta}_T$ change.

As mentioned earlier, in a transfer learning problem we are interested in using both source and target training data to find an estimate $\widehat{\boldsymbol{\theta}}_T$ of $\boldsymbol{\theta}_T$ with small target generalization error. In a minimax framework, $\boldsymbol{\theta}_T$ is chosen in an adversarial way, and the goal is to find an estimate $\hat{\boldsymbol{\theta}}_T$ that achieves the smallest worst case target generalization risk $\sup \mathbb{E}_{\sim \text{ source and target} \atop \text{samples}} \left[ \mathbb{E}_{\mathbb{Q}_{\boldsymbol{\theta}_T}} [\|\boldsymbol{y}_T - \widehat{\boldsymbol{y}}_T\|_{\ell_2}^2] \right]$. Here, the supremum is taken over the class of transfer problems under study (possible $(\mathbb{P}_{\boldsymbol{\theta}_S}, \mathbb{Q}_{\boldsymbol{\theta}_T})$ pairs). We are interested in considering classes of transfer learning problems which properly reflect the difficulty of transfer learning. To this aim we need to have an appropriate notion of similarity or *transfer distance* between source and target tasks. To define the appropriate measure of transfer distance we are guided by the following proposition (see Section **??** for the proof) which characterizes the target generalization error for linear and one-hidden layer neural network models.

**Proposition 1** *Let $\mathbb{Q}_{\boldsymbol{\theta}_T}$ be the data distribution over the target task with parameter $\boldsymbol{\theta}_T$ according to one of the models defined in Section 2.2. The target generalization error of an estimated model with parameter $\widehat{\boldsymbol{\theta}}_T$ is given by:*

- *Linear model:*

$$\mathbb{E}_{\mathbb{Q}_{\boldsymbol{\theta}_T}} [\|\widehat{\boldsymbol{y}}_T - \boldsymbol{y}_T\|_{\ell_2}^2] = \|\boldsymbol{\Sigma}_T^{\frac{1}{2}} (\widehat{\boldsymbol{W}}_T - \boldsymbol{W}_T)^T\|_F^2 + k\sigma^2 \tag{2.2}$$

- *One-hidden layer neural network model with fixed hidden-to-output layer:*

$$\mathbb{E}_{\mathbb{Q}_{\boldsymbol{\theta}_T}} [\|\widehat{\boldsymbol{y}}_T - \boldsymbol{y}_T\|_{\ell_2}^2] \geq \frac{1}{4} \sigma_{min}^2(\boldsymbol{V}) \|\boldsymbol{\Sigma}_T^{\frac{1}{2}} (\widehat{\boldsymbol{W}}_T - \boldsymbol{W}_T)^T\|_F^2 + k\sigma^2 \tag{2.3}$$

- *One-hidden layer neural network model with fixed input-to-hidden layer:*

$$\mathbb{E}_{\mathbb{Q}_{\boldsymbol{\theta}_T}} [\|\widehat{\boldsymbol{y}}_T - \boldsymbol{y}_T\|_{\ell_2}^2] = \|\widetilde{\boldsymbol{\Sigma}}_T^{\frac{1}{2}} (\widehat{\boldsymbol{V}}_T - \boldsymbol{V}_T)^T\|_F^2 + k\sigma^2 \tag{2.4}$$

*Here, $\widetilde{\boldsymbol{\Sigma}}_T := \left[ \frac{1}{2} \|\boldsymbol{a}_i\|_{\ell_2} \|\boldsymbol{a}_j\|_{\ell_2} \frac{\sqrt{1-\gamma_{ij}^2} + (\pi - \cos^{-1}(\gamma_{ij}))\gamma_{ij}}{\pi} \right]_{ij}$ where $\boldsymbol{a}_i$ is the ith row of the matrix $\boldsymbol{W}\boldsymbol{\Sigma}_T^{\frac{1}{2}}$ and $\gamma_{ij} := \frac{\boldsymbol{a}_i^T \boldsymbol{a}_j}{\|\boldsymbol{a}_i\|_{\ell_2} \|\boldsymbol{a}_j\|_{\ell_2}}$.*

Proposition 1 essentially shows how the generalization error is related to an appropriate distance between the estimated and ground truth parameters. This in turn motivates our notion of transfer distance/similarity between source and target tasks discussed next.

**Definition 1** *(Transfer distance) For a source and target task generated according to one of the models in Section 2.2 parametrized by $\boldsymbol{\theta}_S$ and $\boldsymbol{\theta}_T$, we define the transfer distance between these two tasks as follows:*

- *Linear model and one-hidden layer neural network model with fixed hidden-to-output layer:*

$$\rho(\boldsymbol{\theta}_S, \boldsymbol{\theta}_T) = \rho(\boldsymbol{W}_S, \boldsymbol{W}_T) := \|\boldsymbol{\Sigma}_T^{\frac{1}{2}}(\boldsymbol{W}_S - \boldsymbol{W}_T)^T\|_F \tag{2.5}$$

- *One-hidden layer neural network model with fixed input-to-hidden layer:*

$$\rho(\boldsymbol{\theta}_S, \boldsymbol{\theta}_T) = \rho(\boldsymbol{V}_S, \boldsymbol{V}_T) := \|\widetilde{\boldsymbol{\Sigma}}_T^{\frac{1}{2}}(\boldsymbol{V}_S - \boldsymbol{V}_T)^T\|_F \tag{2.6}$$

*where $\widetilde{\Sigma}_T$ is defined in Proposition 1.*

With the notion of transfer distance in hand we are now ready to formally define the class of pairs of distributions over source and target tasks which we focus on in this paper.

**Definition 2** *(Class of pairs of distributions) For a given $\Delta \in \mathbb{R}^+$, $\mathcal{P}_\Delta$ is the class of pairs of distributions over source and target tasks whose transfer distance according to Definition 1 is less than $\Delta$. That is, $\mathcal{P}_\Delta = \{(\mathbb{P}_{\boldsymbol{\theta}_S}, \mathbb{Q}_{\boldsymbol{\theta}_T}) | \rho(\boldsymbol{\theta}_S, \boldsymbol{\theta}_T) \le \Delta\}$.*

With these ingredients in place we are now ready to formally state the transfer learning minimax risk.

$$\mathcal{R}_T(\mathcal{P}_\Delta) := \inf_{\widehat{\boldsymbol{\theta}}_T} \sup_{(\mathbb{P}_{\boldsymbol{\theta}_S}, \mathbb{Q}_{\boldsymbol{\theta}_T}) \in \mathcal{P}_\Delta} \mathbb{E}_{S_{\mathbb{P}_{\boldsymbol{\theta}_S}} \sim \mathbb{P}_{\boldsymbol{\theta}_S}^{1:n_S}} \left[ \mathbb{E}_{S_{\mathbb{Q}_{\boldsymbol{\theta}_T}} \sim \mathbb{Q}_{\boldsymbol{\theta}_T}^{1:n_T}} \left[ \mathbb{E}_{\mathbb{Q}_{\boldsymbol{\theta}_T}} [\|\boldsymbol{y}_T - \widehat{\boldsymbol{y}}_T\|_{\ell_2}^2] \right] \right] \tag{2.7}$$

Here, $S_{\mathbb{P}_{\boldsymbol{\theta}_S}}$ and $S_{\mathbb{Q}_{\boldsymbol{\theta}_T}}$ denote i.i.d. samples $\{(\boldsymbol{x}_S^{(i)}, \boldsymbol{y}_S^{(i)})\}_{i=1}^{n_S}$ and $\{(\boldsymbol{x}_T^{(i)}, \boldsymbol{y}_T^{(i)})\}_{i=1}^{n_T}$ generated from the source and target distributions. We would like to emphasize that $\widehat{y}_T$ as defined in section 2.1, is a function of samples $(S_{\mathbb{P}_{\boldsymbol{\theta}_S}}, S_{\mathbb{Q}_{\boldsymbol{\theta}}})$.

# 3 Main Results

In this section, we provide a lower bound on the transfer learning minimax risk (2.7) for the three transfer learning models defined in Section 2.1. As with any other quantity related to generalization error this risk naturally depends on the size of the model and how correlated the features are in the target model. The following definition aims to capture the effective number of parameters of the model.

**Definition 3** *(Effective dimension) The effective dimension of the three models defined in Section 2.1 are defined as follows:*

- *Linear model: $D := rank(\boldsymbol{\Sigma}_T)k - 1$,*

- *One-hidden layer neural network model with fixed hidden-to-output layer: $D := rank(\boldsymbol{\Sigma}_T)\ell - 1$,*

- *One-hidden layer neural network model with fixed input-to-hidden layer: $D := rank(\widetilde{\boldsymbol{\Sigma}}_T)k - 1$.*

Our results also depend on another quantity which we refer to as the transfer coefficient. Roughly speaking these quantities are meant to capture the relative effectiveness of a source training data from the perspective of the generalization error of the target task and vice versa.

**Definition 4** *(Transfer coefficients) Let $n_S$ and $n_T$ be the number of source and target training data. We define the transfer coefficients in the three models defined in Section 2.1 as follows*

- *Linear model: $r_S := \left\| \boldsymbol{\Sigma}_S^{\frac{1}{2}} \boldsymbol{\Sigma}_T^{-\frac{1}{2}} \right\|^2$ and $r_T := 1$.*

- *One-hidden layer neural net with fixed output layer: $r_S := \left\| \boldsymbol{\Sigma}_S^{\frac{1}{2}} \boldsymbol{\Sigma}_T^{-\frac{1}{2}} \right\|^2 \|\boldsymbol{V}\|^2$ and $r_T := \|\boldsymbol{V}\|^2$.*

- *One-hidden layer neural net model with fixed input layer: $r_S := \left\| \widetilde{\boldsymbol{\Sigma}}_S^{\frac{1}{2}} \widetilde{\boldsymbol{\Sigma}}_T^{-\frac{1}{2}} \right\|^2$ and $r_T := 1$. Here,*

$\widetilde{\boldsymbol{\Sigma}}_S := \left[ \frac{1}{2} \|\boldsymbol{c}_i\|_{\ell_2} \|\boldsymbol{c}_j\|_{\ell_2} \frac{\sqrt{1-\widetilde{\gamma}_{ij}^2} + (\pi - \cos^{-1}(\widetilde{\gamma}_{ij}))\widetilde{\gamma}_{ij}}{\pi} \right]_{ij}$ *where $\boldsymbol{c}_i$ is the $i$th row of $\boldsymbol{W}\boldsymbol{\Sigma}_S^{\frac{1}{2}}$ and $\widetilde{\gamma}_{ij} = \frac{\boldsymbol{c}_i^T \boldsymbol{c}_j}{\|\boldsymbol{c}_i\|_{\ell_2} \|\boldsymbol{c}_j\|_{\ell_2}}$ and $\widetilde{\boldsymbol{\Sigma}}_T$ are defined per Proposition 1.*

*In the above expressions $\|\cdot\|$ stands for the operator norm. Furthermore, we define the effective number of source and target samples as $r_S n_S$ and $r_T n_T$, respectively.*

With these definitions in place we now present our lower bounds on the transfer learning minimax risk of any algorithm for the linear and one-hidden layer neural network models (see the supplementary material for the proof).

**Theorem 1** *Consider the three transfer learning models defined in Section 2.1 consisting of $n_S$ and $n_T$ source and target training data generated i.i.d. according to a class of source/target distributions with transfer distance at most $\Delta$ per Definition 2. Moreover, let $r_S$ and $r_T$ be the source and target transfer coefficients per Definition 4. Furthermore, assume the effective dimension $D$ per Definition 3 obeys $D \geq 20$. Then, the transfer learning minimax risk (2.7) obeys the following lower bounds:*

- *Linear model: $\mathcal{R}_T(\mathcal{P}_\Delta) \geq B + k\sigma^2$.*

- *One-hidden layer neural network with fixed hidden-to-output layer: $\mathcal{R}_T(\mathcal{P}_\Delta) \geq \frac{1}{4}\sigma_{min}^2(\boldsymbol{V})B + k\sigma^2$.*

- *One-hidden layer neural network model with fixed input-to-hidden layer: $\mathcal{R}_T(\mathcal{P}_\Delta) \geq B + k\sigma^2$.*

*Here, $\sigma_{min}(\boldsymbol{V})$ denotes the minimum singular value of $\boldsymbol{V}$ and*

$$B := \begin{cases} \frac{\sigma^2 D}{256 r_T n_T}, & \text{if } \Delta \geq \sqrt{\frac{\sigma^2 D \log 2}{r_T n_T}} \\ \frac{1}{100}\Delta^2\left[1 - 0.8\frac{r_T n_T \Delta^2}{\sigma^2 D}\right], & \text{if } \frac{1}{45}\sqrt{\frac{\sigma^2 D}{r_S n_S + r_T n_T}} \leq \Delta < \sqrt{\frac{\sigma^2 D \log 2}{r_T n_T}} \\ \frac{\Delta^2}{1000} + \frac{6}{1000}\frac{D\sigma^2}{r_S n_S + r_T n_T}, & \text{if } \Delta < \frac{1}{45}\sqrt{\frac{\sigma^2 D}{r_S n_S + r_T n_T}} \end{cases} \tag{3.1}$$

Note that, the nature of the lower bound and final conclusions provided by the above theorem are similar for all three models. More specifically, Theorem 1 leads to the following conclusions:

- **Large transfer distance** ($\Delta \geq \sqrt{\frac{D\sigma^2 \log 2}{r_T n_T}}$). When the transfer distance between the source and target tasks is large, source samples are helpful in decreasing the target generalization error until the error reaches $\frac{\sigma^2 D}{256 r_T n_T}$. Beyond this point, by increasing the number of source samples, target generalization error does not decrease further and it becomes dominated by the target samples. In other words, when the distance is large, source samples cannot compensate for target samples.

- **Moderate distance** ($\frac{1}{45}\sqrt{\frac{\sigma^2 D}{r_S n_S + r_T n_T}} \leq \Delta < \sqrt{\frac{\sigma^2 D \log 2}{r_T n_T}}$). The lower bound of this regime suggests that if the distance between the source and target tasks is strictly positive, i.e $\Delta > 0$, even if we have infinitely many source samples, target generalization error still does not go to zero and depends on the number of available target samples. In other words, source samples cannot compensate for the lack of target samples.

- **Small distance** ($\Delta < \frac{1}{45}\sqrt{\frac{\sigma^2 D}{r_S n_S + r_T n_T}}$). In this case, the lower bound on the target generalization error scales with $\frac{1}{r_S n_S + n_T r_T}$ where $r_S n_S$ and $r_T n_T$ are the effective number of source and target samples per Definition 4. Hence, when $\Delta$ is small, the target generalization error scales with the reciprocal of the total effective number of source and target samples which means that source samples are indeed helpful in reducing the target generalization error and every source sample is roughly equivalent to $\frac{r_S}{r_T}$ target samples. Furthermore, when the distance of source and target is zero, i.e. $\Delta = 0$, the lower bound reduces to $\frac{6}{1000}\frac{D\sigma^2}{r_S n_S + r_T n_T}$. Conforming with our intuition, in this case the bound resembles a non-transfer learning scenario where a combination of source and target samples are used. Indeed, the lower bound is proportional to the noise level, effective dimension and the total number of samples matching typical statistical learning lower bounds.

## 4 Experiments and Numerical Results

We demonstrate the validity of our theoretical framework through experiments on real datasets sampled from ImageNet as well as synthetic simulated data. The experiments on ImageNet data allow us to investigate the impact of transfer distance and noise parameters appearing in Theorem 1 on the target generalization error. However, since the source and target tasks are both image classification, they are inherently correlated with each other and we cannot expect a wide range of transfer distances between them. Therefore, we carry out a more in-depth study on simulated data to investigate the effect of the number of source and target samples on the target generalization error in different transfer distance regimes. Full source code to reproduce the results is available at (16).

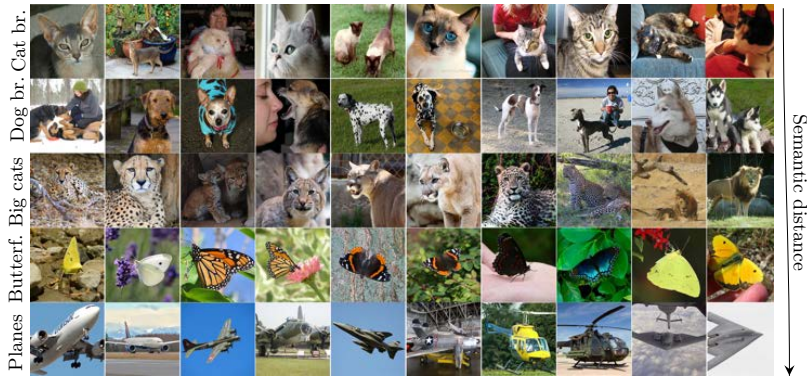

Figure 1: Sample images from the source/target datasets derived from ImageNet. Transfer distance increases from top to bottom.

| Source / target task | $\rho(source, target)$ | Validation loss | Noise level ($\sigma$) |
|---|---|---|---|
| *cat breeds / dog breeds* | 11.62 | 0.2194 | 0.2095 |
| *cat breeds / big cats* | 12.35 | 0.1682 | 0.1834 |
| *cat breeds / butterflies* | 13.48 | 0.1367 | 0.1653 |
| *cat breeds / planes* | 16.41 | 0.1450 | 0.1703 |

Table 1: Transfer distance and noise level for various source-target pairs.

## 4.1 ImageNet Experiments

Here we verify our theoretical formulation on a subset of ImageNet, a well-known image classification dataset and show that our main theorem conforms with practical transfer learning scenarios.

**Sample datasets.** We create five datasets by sub-sampling 2000 images in five classes from ImageNet (400 examples per class). As depicted in Figure 1, we deliberately compile datasets covering a spectrum of semantic distances from each other in order to study the utility/effect of transfer distance on transfer learning. The picked datasets are as follows: *cat breeds*, *big cats*, *dog breeds*, *butterflies*, *planes*. For details of the classes in each dataset please refer to (16). We pass the images through a VGG16 network pretrained on ImageNet with the fully connected top classifier removed and use the extracted features instead of the actual images. We set aside 10% of the dataset as test set. Furthermore, 10% of the remaining data is used for validation and 90% for training. In the following we fix identifying *cat breeds* as the source task and the four other datasets as target tasks.

**Training.** We trained a one-hidden layer neural network for each dataset. To facilitate fair comparison between the trained models in weight-space, we fixed a random initialization of the hidden-to-output layer shared between all networks and we only trained over the input-to-hidden layer (in accordance with the theoretical formulation). Moreover, we used the same initialization of input-to-hidden weights. We trained a separate model on each of the five datasets on MSE loss with one-hot encoded labels. We use an Adam optimizer with a learning rate of 0.0001 and train for 100 epochs or until the network reaches 99.5% accuracy whichever occurs first. The target noise levels are calculated based on the average loss of the trained ground truth models on the target validation set (note that this average loss equals $k\sigma^2 = 5\sigma^2$).

**Results.** First, we calculate the transfer distance from Definition 1 between the model trained on the source task (*cat breeds*) and the other four models trained on target tasks by fitting a ground truth model to each task using complete training data. Our results depicted in Table 1 demonstrate that the introduced metric strongly correlates with perceived semantic distance. The closest tasks, *cat breeds* and *dog breeds*, are both characterized by pets with similar features, and with humans frequently appearing in the images. Images in the second closest pair, *cat breeds* and *big cats*, include animals with similar features, but *big cats* have more natural scenes and less humans compared with *dog breeds*, resulting in slightly higher distance from the source task. As expected, *cat breeds-butterflies* distance is significantly higher than in case of the previous two targets, but they share some characteristics such as the presence of natural backgrounds. The largest distance is between *cat breeds* and *planes*, which is clearly the furthest task semantically as well.

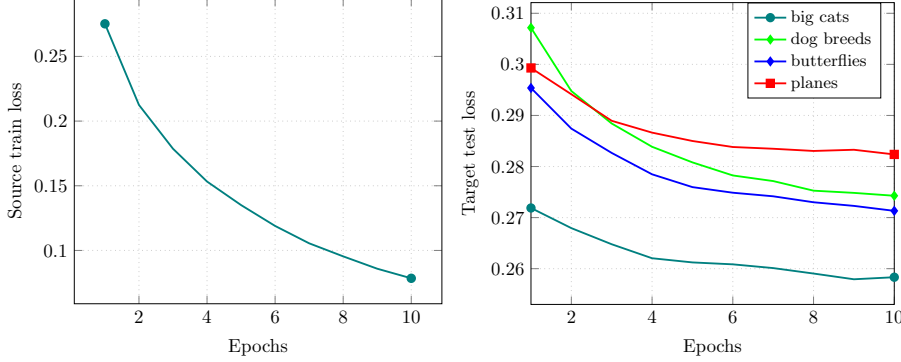

Figure 2: Train and test loss of a one-hidden layer network trained on *cat breeds* dataset.

Our next set of experiments focuses on checking whether the transfer distance is indicative of transfer target risk/generalization error. To this aim we use a very simple transfer learning approach where we use only source data to train a one-hidden layer network as described before and measure its performance on the target tasks. Note that the network has never seen examples of the target dataset. Figure 2 depicts how train and test loss evolved over the training process. We stop after 10 epochs when validation losses on target tasks have more or less stabilized. The results closely match our expectations from Theorem 1. Based on Table 1 the noise level of ground truth models for *big cats*, *butterflies* and *planes* are about the same and therefore their test loss follows the same ordering as their distances from the source task (see Table 1). Moreover, even though *dog breeds* has the lowest distance from the source task, it is also the noisiest. The lower bound in Theorem 1 includes an additive noise term, and therefore the change in ordering between *dog breeds* and *butterflies* is justified by our theory and demonstrates the effect of the target task noise level on generalization.

**Theoretical lower bounds for the ImageNet experiments.** In order to plot the theoretical lower bounds, first we estimate the parameters appearing in the bounds. Then using those parameters we depict the lower bounds in Figure 3. In Figure 3 each plot consists of two lower bounds, namely a crude bound (presented in Theorem 1) and a more precise bound presented in the proofs.

## 4.2   Numerical Results

In this section we perform synthetic numerical simulations in order to carefully cover all regimes of transfer distance from our main theorem, and show how the target generalization error depends on the number of source and target samples in different regimes.

**Experimental setup 1.** First, we generate data according to the linear model with parameters $d = 200, k = 30, \sigma = 1, \mathbf{\Sigma}_S = 2 \cdot I_d, \mathbf{\Sigma}_T = I_d$. Then we generate the source parameter matrix $\mathbf{W}_S \in \mathbb{R}^{k \times d}$ with elements sampled from $\mathcal{N}(0, 10)$. Furthermore, we generate two target parameter matrices $\mathbf{W}_{T_1}$ and $\mathbf{W}_{T_2} \in \mathbb{R}^{k \times d}$ for tasks $T_1$ and $T_2$ such that $\mathbf{W}_{T_1} = \mathbf{W}_S + \mathbf{M}_1$ and $\mathbf{W}_{T_2} = \mathbf{W}_S + \mathbf{M}_2$ where the elements of $\mathbf{M}_1, \mathbf{M}_2$ are sampled from $\mathcal{N}(0, 10^{-3})$ and $\mathcal{N}(0, 3.6 \times 10^5)$ respectively. Similarly for the one-hidden layer neural network model when the the output layer is fixed, we set the parameters $k = 1, \ell = 30, d = 200, \sigma = 1, \mathbf{\Sigma}_S = 2 \cdot I_d, \mathbf{\Sigma}_T = I_d$ and $V = \mathbf{1}_{k \times \ell}$. We also use the same $\mathbf{W}_S, \mathbf{W}_{T_1}, \mathbf{W}_{T_2}$ as in the linear model. We note that the transfer distance between the source task to target task $T_1$ is small but the transfer distance between the source task to target task $T_2$ is large ($\rho(\mathbf{W}_S, \mathbf{W}_{T_1}) = .0183$ and $\rho(\mathbf{W}_S, \mathbf{W}_{T_2}) = 116.694$).

**Training approach 1.** We test the performance of a simple transfer learning approach. Given $n_S$ source samples and $n_T$ target samples, we estimate $\widehat{\mathbf{W}}_T$ by minimizing the weighted empirical risk

$$\min_{\mathbf{W}} \frac{1}{2n_T} \sum_{i=1}^{n_T} \left\| f(\mathbf{W}; \mathbf{x}_T^{(i)}) - \mathbf{y}_T^{(i)} \right\|_{\ell_2}^2 + \frac{\lambda}{2n_S} \sum_{j=1}^{n_S} \left\| f(\mathbf{W}; \mathbf{x}_S^{(j)}) - \mathbf{y}_S^{(j)} \right\|_{\ell_2}^2 \tag{4.1}$$

We then evaluate the generalization error by testing the estimated model $\widehat{\mathbf{W}}_T$ on 200 unseen test data points generated by the target model. All reported plots are the average of 10 trials.

**Results 1.** Figure 4 (a) depicts the target generalization error for target tasks $T_1$ and $T_2$ for the linear model for different $n_S$ values with $\lambda = 1$ and $n_T = 50$. Figure 4 (b) depicts the target generalization error for tasks $T_1$ and target $T_2$ for the linear model for different $n_T$ values with the number of source

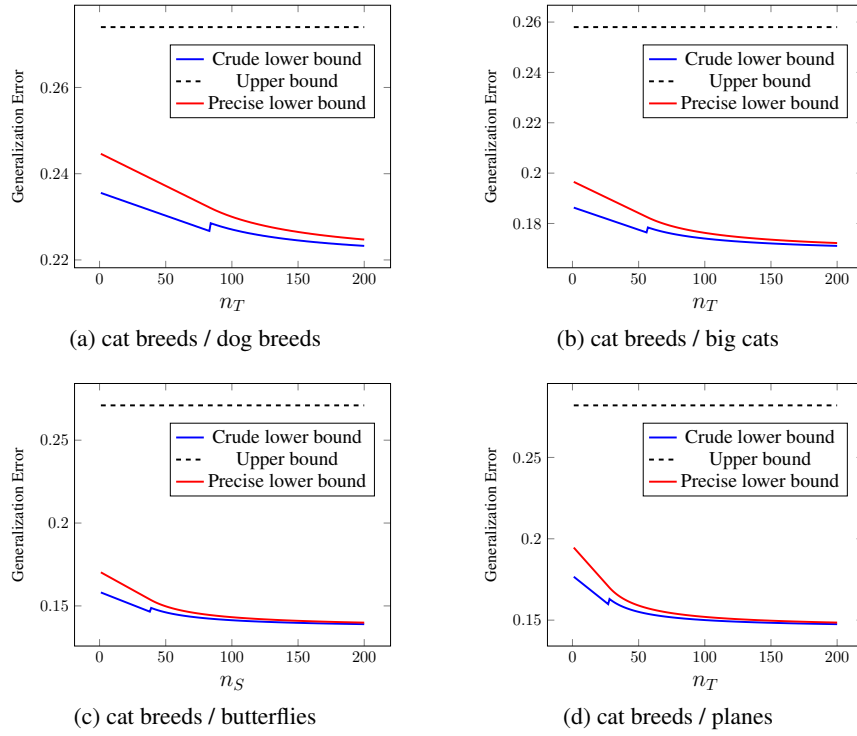

Figure 3: Theoretical lower bounds and experimental upper bounds.

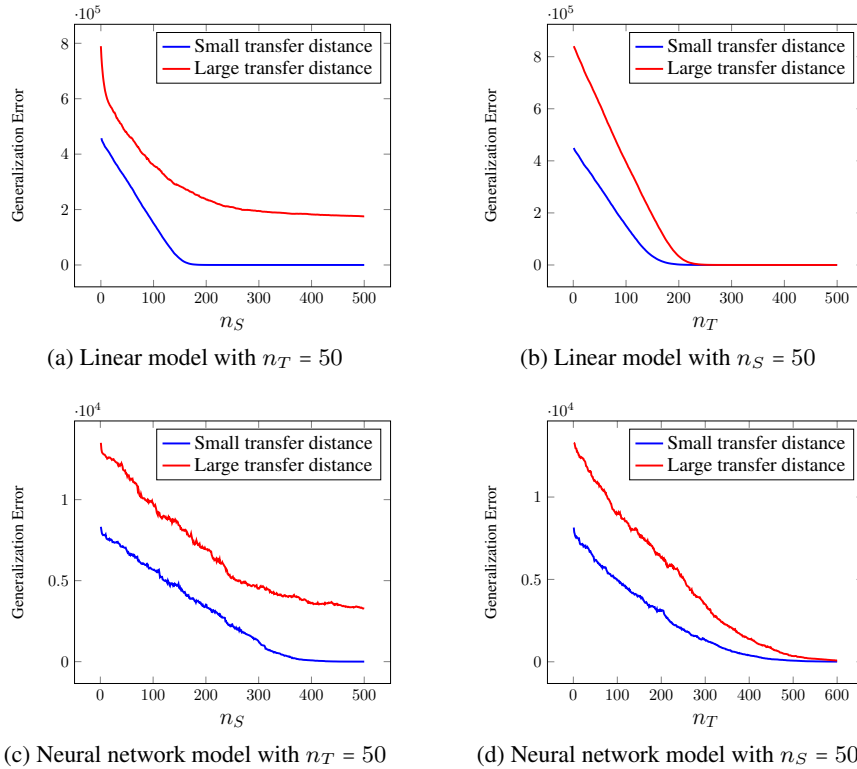

Figure 4: Target generalization error for a linear model ((a) and (b)) and a neural network model with fixed hidden-to-output layer ((c) and (d)).

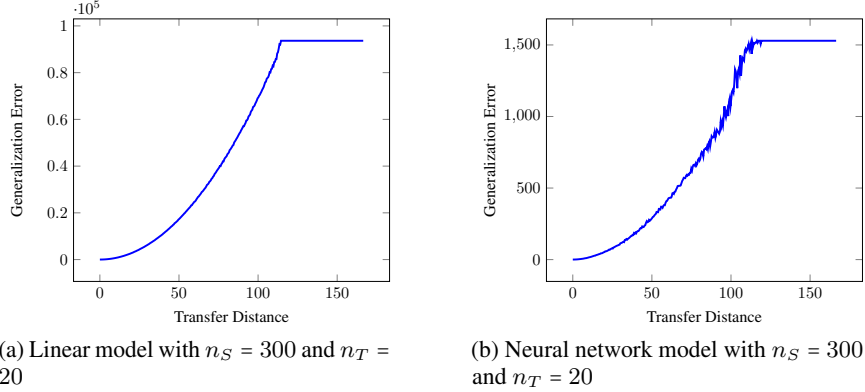

(a) Linear model with $n_S = 300$ and $n_T = 20$

(b) Neural network model with $n_S = 300$ and $n_T = 20$

Figure 5: Target generalization error for a linear model (a) and a neural network model with fixed hidden-to-output layer (b).

samples fixed at $n_S = 50$. Here, we set $\lambda = 1$ for target task $T_1$, where the transfer distance from source is small, and $\lambda = .001$ for target task $T_2$, where the transfer distance from source is large. Figures 4 (c) and 4 (d) have the same settings as in Figures 4 (a) and 4 (b) but we use a one-hidden layer neural network model with fixed hidden-to-output weights in lieu of the linear model.

Figures 4 (a) and (c) clearly demonstrate that when the transfer distance between the source and target tasks is large, increasing the number of source samples is not helpful beyond a certain point. In particular, the target generalization error starts to saturate and does not decrease further. Stated differently, in this case the source samples cannot compensate for the target samples. This observation conforms with our main theoretical result. Indeed, when the transfer distance $\Delta$ is large, $B$ is lower bounded by $\frac{\sigma^2 D}{256 r_T n_T}$ which is independent of the number of source samples $n_S$. Furthermore, these figures also demonstrate that when the transfer distance is small, increasing the number of source samples is helpful and results in lower target generalization error. This also matches our theoretical results as when the transfer distance $\Delta$ is small, the target generalization error is proportional to $\frac{D\sigma^2}{r_S n_S + r_T n_T}$.

Figures 4 (b) and (d) indicate that regardless of the transfer distance between the source and target tasks the target generalization error steadily decreases as the number of target samples increases. This is a good match with our theoretical results as $n_T$ appears in the denominator of our lower bound in all three regimes.

To further investigate the effect of transfer distance between the source and target on the target generalization error we consider another set of experiments below.

**Experimental setup 2.** For the linear model, we use the parameters $d = 50, k = 30, \sigma = 0.3, \boldsymbol{\Sigma}_S = 2 \cdot \boldsymbol{I}_d$, and $\boldsymbol{\Sigma}_T = \boldsymbol{I}_d$. We generate the target parameter $\boldsymbol{W}_T \in \mathbb{R}^{k \times d}$ with entries generated i.i.d. $\mathcal{N}(0, 10)$. To create different transfer distances between the source and target data we then generate the source parameter $\boldsymbol{W}_S \in \mathbb{R}^{k \times d}$ as $\boldsymbol{W}_S = \boldsymbol{W}_T + i \cdot \boldsymbol{M}$ where the elements of the matrix $\boldsymbol{M}$ are sampled from $\mathcal{N}(0, 10^{-4})$ and $i$ varies between 1 and $140000$ in increments of $400$. Similarly for the one-hidden layer neural network model when the the output layer is fixed, we pick parameter values $k = 1, \ell = 30, d = 50, \sigma = 0.3, \boldsymbol{\Sigma}_S = 2 \cdot \boldsymbol{I}_d$, and $\boldsymbol{\Sigma}_T = \boldsymbol{I}_d$ and set all of the entries of $V$ equal to one. Furthermore, we use the same source and target parameters $\boldsymbol{W}_S$ and $\boldsymbol{W}_T$ as in the linear model.

**Training approach 2.** Given $n_S = 300$ and $n_T = 20$ source and target samples we minimize the weighted empirical risk (4.1). In this experiment we pick $\lambda \in \{0, \frac{1}{4}, \frac{1}{2}, \frac{3}{4}, 1\}$ that minimizes a validation set consisting of 50 data points created from the same distribution as the target task. Finally we test the estimated model on 200 unseen target test data points. The reported numbers are based on an average of 20 trials .

**Results 2.** Fig. 5 depicts the target generalization error as a function of the transfer distance between the source and target in the linear and neural network models. This figure clearly shows that when the transfer distance is small, the generalization error has a quadratic growth. However, as the distance increases the error saturates which matches the behavior of $\Delta$ predicted by our lower bounds.

## Broader Impact

While our work is theoretical/foundations in nature let us discuss a few ways in which it may have broader impacts. In this paper, we characterize a lower bound for transfer learning in the context of linear models and one-hidden layer neural networks. More specifically, we provide a lower bound for target generalization error in terms of the number of source and target tasks and an appropriately defined transfer distance between the source and target tasks. Given the amount of effort dedicated to data collection, curation, and storage, a precise understanding of the amount of data needed may help utilize a variety of resources more effectively. Moreover, our results may guide practitioners to when there is no hope of knowledge transfer from one domain to another. This may help avoid unwarranted generalizations from one situation/environment to unrelated instances. On the other hand, it is worth emphasizing that this paper focuses on shallow linear/neural network models and does not capture more realistic Deep Neural Network (DNN) models typically used in practice. Therefore, one has to be cautious in over-interpreting the results of this paper for general DNN models.

## 5 Acknowledgments and Disclosure of Funding

This material is based upon work supported by Defense Advanced Research Projects Agency (DARPA) under Contract No. FA8750-19-2-1005. The views, opinions, and/or findings expressed are those of the author(s) and should not be interpreted as representing the official views or policies of the Department of Defense or the U.S. Government.

M. Soltanolkotabi is also supported by the Packard Fellowship in Science and Engineering, a Sloan Research Fellowship in Mathematics, an NSF-CAREER under award #1846369, the Air Force Office of Scientific Research Young Investigator Program (AFOSR-YIP) under award #FA9550 − 18 − 1 − 0078, DARPA FastNICS program, and NSF-CIF awards #1813877 and #2008443.

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
