[Supplementary Material]

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

 $W_{T_1} = W_S + M_1$ and $W_{T_2} = W_S + M_2$ where the elements of $M_1, M_2$ are sampled from $\mathcal{N}(0, 10^{-3})$ and $\mathcal{N}(0, 3.6 \times 10^5)$ respectively. Similarly for the one-hidden layer neural network model when the the output layer is fixed, we set the parameters $k = 1, \ell = 30, d = 200, \sigma = 1, \Sigma_S = 2 \cdot I_d, \Sigma_T = I_d$ and $V = \mathbf{1}_{k \times \ell}$. We also use the same $W_S, W_{T_1}, W_{T_2}$ as in the linear model. We note that the transfer distance between the source task to target task $T_1$ is small but the transfer distance between the source task to target task $T_2$ is large $(\rho(W_S, W_{T_1}) = .0183$ and $\rho(W_S, W_{T_2}) = 116.694)$.

**Training approach 1.** We test the performance of a simple transfer learning approach. Given $n_S$ source samples and $n_T$ target samples, we estimate $\widehat{W}_T$ by minimizing the weighted empirical risk

$$\min_{W} \frac{1}{2n_T} \sum_{i=1}^{n_T} \left\| f(W; x_T^{(i)}) - y_T^{(i)} \right\|_{\ell_2}^2 + \frac{\lambda}{2n_S} \sum_{j=1}^{n_S} \left\| f(W; x_S^{(j)}) - y_S^{(j)} \right\|_{\ell_2}^2 \tag{4.1}$$

We then evaluate the generalization error by testing the estimated model $\widehat{W}_T$ on 200 unseen test data points generated by the target model. All reported plots are the average of 10 trials.

**Results 1.** Figure 4 (a) depicts the target generalization error for target tasks $T_1$ and $T_2$ for the linear model for different $n_S$ values with $\lambda = 1$ and $n_T = 50$. Figure 4 (b) depicts the target generalization error for tasks $T_1$ and target $T_2$ for the linear model for different $n_T$ values with the number of source

(a) cat breeds / dog breeds

(b) cat breeds / big cats

(c) cat breeds / butterflies

(d) cat breeds / planes

Figure 3: Theoretical lower bounds and experimental upper bounds.

(a) Linear model with $n_T = 50$

(b) Linear model with $n_S = 50$

(c) Neural network model with $n_T = 50$

(d) Neural network model with $n_S = 50$

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

# 6 Proof outline and proof of Theorem 1 in the linear model

In this section we present a sketch of the proof of Theorem 1 for the linear model. The proof for the neural network models follow a similar approach and appear in Sections 7.5 and 7.7.

Note that by Proposition 1, the generalization error is given by

$$\mathbb{E}_{\mathbb{Q}_{\boldsymbol{\theta}_T}}\left[\|\widehat{\boldsymbol{y}}_T - \boldsymbol{y}_T\|_{\ell_2}^2\right] = \|\boldsymbol{\Sigma}_T^{\frac{1}{2}}(\widehat{\boldsymbol{W}}_T - \boldsymbol{W}_T)^T\|_F^2 + k\sigma^2.$$

Therefore, in order to find a minimax lower bound on the target generalization error, it suffices to find a lower bound for the following quantity

$$\mathcal{R}_T(\mathcal{P}_\Delta; \phi \circ \rho)$$
$$:= \inf_{\widehat{\boldsymbol{W}}_T} \sup_{(\mathbb{P}_{\boldsymbol{W}_S}, \mathbb{Q}_{\boldsymbol{W}_T}) \in \mathcal{P}_\Delta} \mathbb{E}_{S_{\mathbb{P}_{\boldsymbol{W}_S}} \sim \mathbb{P}_{\boldsymbol{W}_S}^{1:n_\mathbb{P}}}\left[\mathbb{E}_{S_{\mathbb{Q}_{\boldsymbol{W}_T}} \sim \mathbb{Q}_{\boldsymbol{W}_T}^{1:n_\mathbb{Q}}}\left[\phi(\rho(\widehat{\boldsymbol{W}}_T(S_{\mathbb{P}_{\boldsymbol{W}_S}}, S_{\mathbb{Q}_{\boldsymbol{W}_T}}), \boldsymbol{W}_T))\right]\right]$$

(6.1)

where $\phi(x) = x^2$ for $x \in \mathbb{R}$ and $\rho$ is per Definition 1. By using well-known techniques from the statistical minimax literature we reduce the problem of finding a lower bound to a hypothesis testing problem (e.g. see (17, Chapter 15)). Since we are estimating the target parameter, i.e. $\boldsymbol{W}_T$, to apply this framework we need to pick $N$ pairs of distributions from the set $\mathcal{P}_\Delta$ such that their target parameters are $2\delta$-separated by the transfer distance per Definition 1. To be more precise, we pick $N$ arbitrary pairs of distributions from $\mathcal{P}_\Delta$ :

$$(\mathbb{P}_{\boldsymbol{W}_S^{(1)}}, \mathbb{Q}_{\boldsymbol{W}_T^{(1)}}), ..., (\mathbb{P}_{\boldsymbol{W}_S^{(N)}}, \mathbb{Q}_{\boldsymbol{W}_T^{(N)}})$$

such that

$$\rho(\boldsymbol{W}_T^{(i)}, \boldsymbol{W}_T^{(j)}) \geq 2\delta \text{ for each } i \neq j \in [N] \times [N] \text{ } (2\delta\text{-separated set})$$

and

$$\rho(\boldsymbol{W}_S^{(i)}, \boldsymbol{W}_T^{(i)}) \leq \Delta \text{ for each } i \in [N] \text{ } (\text{as they belong to } \mathcal{P}_\Delta)$$

With these $N$ distribution pairs in place we can follow a proof similar to that of (17, Proposition 15.1) to reduce the minimax problem to a hypothesis test problem. In particular, consider the following $N$-array hypothesis testing problem:

- $J$ is the uniform distribution over the index set $[N] := \{1, 2, ..., N\}$
- Given $J = i$, generate $n_S$ i.i.d. samples from $\mathbb{P}_{\boldsymbol{W}_S^{(i)}}$ and $n_T$ i.i.d. samples from $\mathbb{Q}_{\boldsymbol{W}_T^{(i)}}$.

Here the goal is to find the true index using $n_S + n_T$ available samples by a testing function $\psi$ from the samples to the indices.

Let $E$ and $F$ be random variables such that $E|\{J = i\} \sim \mathbb{P}_{\boldsymbol{W}_S^{(i)}}$ and $F|\{J = i\} \sim \mathbb{Q}_{\boldsymbol{W}_T^{(i)}}$. Furthermore, let $Z_\mathbb{P}$ and $Z_\mathbb{Q}$ consist of $n_S$ independent copies of random variable $E$ and $n_T$ independent copies of random variable $F$, respectively. In this setting, by slightly modifying the (17, Proposition 15.1) we can conclude that

$$\mathcal{R}_T(\mathcal{P}_\Delta; \phi \circ \rho) \geq \phi(\delta) \frac{1}{N} \sum_{i=1}^{N} \text{Prob}(\psi(Z_\mathbb{P}, Z_\mathbb{Q}) \neq i)$$

where

$$\psi(Z_\mathbb{P}, Z_\mathbb{Q}) := \arg\min_{n \in [N]} \rho(\widehat{\boldsymbol{W}}_T, \boldsymbol{W}_T^n).$$

Furthermore, by using Fano's inequality we can conclude that

$$\mathcal{R}_T(\mathcal{P}_\Delta; \phi \circ \rho) \geq \phi(\delta) \frac{1}{N} \sum_{i=1}^{N} \mathbb{P}\left\{\psi(Z_\mathbb{P}, Z_\mathbb{Q}) \neq i\right\}$$

$$\geq \phi(\delta)\left(1 - \frac{I(J; (Z_\mathbb{P}, Z_\mathbb{Q})) + \log 2}{\log N}\right)$$

$$\geq \phi(\delta)\left(1 - \frac{I(J; Z_\mathbb{P}) + I(J; Z_\mathbb{Q}) + \log 2}{\log N}\right)$$

$$\geq \phi(\delta)\left(1 - \frac{n_S I(J; E) + n_T I(J; F) + \log 2}{\log N}\right).$$

(6.2)

Here the third inequality is due to the fact that given $J = i$, $Z_\mathbb{P}$ and $Z_\mathbb{Q}$ are independent. To continue further, note that we can bound the mutual information by the following KL-divergences

$$I(J; E) \le \frac{1}{N^2} \sum_{i,j} D_{KL}(\mathbb{P}_{\boldsymbol{W}_S^{(i)}} \| \mathbb{P}_{\boldsymbol{W}_S^{(j)}})$$

$$I(J; F) \le \frac{1}{N^2} \sum_{i,t} D_{KL}(\mathbb{Q}_{\boldsymbol{W}_T^{(i)}} \| \mathbb{Q}_{\boldsymbol{W}_T^{(j)}}). \tag{6.3}$$

In the next lemma, proven in Section 7.2, we explicitly calculate the above KL-divergences.

**Lemma 1** *Suppose that $\mathbb{P}_{\boldsymbol{W}_S^{(i)}}$ and $\mathbb{P}_{\boldsymbol{W}_S^{(j)}}$ are the joint distributions of features and labels in a source task and $\mathbb{Q}_{\boldsymbol{W}_T^{(i)}}$ and $\mathbb{Q}_{\boldsymbol{W}_T^{(j)}}$ are joint distributions of features and labels in a target task as defined in Section 2.1 for the linear model. Then $D_{KL}(\mathbb{P}_{\boldsymbol{W}_S^{(i)}} \| \mathbb{P}_{\boldsymbol{W}_S^{(j)}}) = \frac{\|\boldsymbol{\Sigma}_S^{\frac{1}{2}}(\boldsymbol{W}_S^{(i)} - \boldsymbol{W}_S^{(j)})^T\|_F^2}{2\sigma^2}$ and $D_{KL}(\mathbb{Q}_{\boldsymbol{W}_T^{(i)}} \| \mathbb{Q}_{\boldsymbol{W}_T^{(j)}}) = \frac{\|\boldsymbol{\Sigma}_T^{\frac{1}{2}}(\boldsymbol{W}_T^{(i)} - \boldsymbol{W}_T^{(j)})^T\|_F^2}{2\sigma^2}$.*

In the following two lemmas, we use local packing techniques to further simplify (6.2) using (6.3) and find minimax lower bounds in different transfer distance regimes. We defer the proof of these lemmas to Sections 7.3 and 7.4.

**Lemma 2** *Assume $\Delta \ge \sqrt{\frac{\sigma^2 D \log 2}{r_T n_T}}$, where $n_T$ is the number of target samples and $D$ and $r_T$ are defined per Definitions 3 and 4. Then we have the following lowerbound*

$$\mathcal{R}_T(\mathcal{P}_\Delta; \phi \circ \rho) \ge \frac{\sigma^2 D}{256 r_T n_T}. \tag{6.4}$$

*Furthermore, if $\Delta < \sqrt{\frac{\sigma^2 D \log 2}{r_T n_T}}$ then*

$$\mathcal{R}_T(\mathcal{P}_\Delta; \phi \circ \rho) \ge \frac{1}{100} \Delta^2 \left(1 - 0.8 \frac{r_T n_T \Delta^2}{\sigma^2 D}\right). \tag{6.5}$$

**Lemma 3** *Assume we have access to $n_S$ source samples as well as $n_T$ target samples and the transfer distance obeys $\Delta \le \frac{1}{45} \sqrt{\frac{\sigma^2 D}{r_S n_S + r_T n_T}}$, where $D$, $r_S$, and $r_T$ are per Definitions 3 and 4. Then,*

$$\mathcal{R}_T(\mathcal{P}_\Delta; \phi \circ \rho) \ge \frac{\Delta^2}{1000} + \frac{6}{1000} \frac{D\sigma^2}{r_S n_S + r_T n_T}. \tag{6.6}$$

The proof of the lower bound in Theorem 1 is complete by combining Lemmas 2 and 3.

# 7 Appendix A

## 7.1 Calculating the Generalization Errors ( Proof of Proposition 1)

- **Linear model:**

By expanding the expression we get

$$\begin{aligned}
\mathbb{E}_{\mathbb{Q}_{\boldsymbol{\theta}_T}}[\|\widehat{\boldsymbol{y}}_T - \boldsymbol{y}_T\|_{\ell_2}^2] &= \mathbb{E}[\|\widehat{\boldsymbol{W}}_T \boldsymbol{x}_T - \boldsymbol{W}_T \boldsymbol{x}_T - w_T\|_{\ell_2}^2] \\
&= \mathbb{E}[\|\widehat{\boldsymbol{W}}_T \boldsymbol{x}_T - \boldsymbol{W}_T \boldsymbol{x}_T\|_{\ell_2}^2] + k\sigma^2 \\
&= \mathbb{E}[\boldsymbol{x}_T^T (\boldsymbol{W}_T - \widehat{\boldsymbol{W}_T})^T (\boldsymbol{W}_T - \widehat{\boldsymbol{W}_T}) \boldsymbol{x}_T] + k\sigma^2 \\
&= \mathbb{E}[\text{trace}(\boldsymbol{x}_T^T (\boldsymbol{W}_T - \widehat{\boldsymbol{W}_T})^T (\boldsymbol{W}_T - \widehat{\boldsymbol{W}_T}) \boldsymbol{x}_T)] + k\sigma^2 \\
&= \mathbb{E}[\text{trace}((\boldsymbol{W}_T - \widehat{\boldsymbol{W}_T})^T (\boldsymbol{W}_T - \widehat{\boldsymbol{W}_T}) \boldsymbol{x}_T \boldsymbol{x}_T^T)] + k\sigma^2 \\
&= \text{trace}((\boldsymbol{W}_T - \widehat{\boldsymbol{W}_T})^T (\boldsymbol{W}_T - \widehat{\boldsymbol{W}_T}) \mathbb{E}[\boldsymbol{x}_T \boldsymbol{x}_T^T]) + k\sigma^2 \\
&= \text{trace}((\boldsymbol{W}_T - \widehat{\boldsymbol{W}_T})^T (\boldsymbol{W}_T - \widehat{\boldsymbol{W}_T}) \boldsymbol{\Sigma}_T) + k\sigma^2 \\
&= \|\boldsymbol{\Sigma}_T^{\frac{1}{2}} (\boldsymbol{W}_T - \widehat{\boldsymbol{W}_T})^T\|_F^2 + k\sigma^2 \tag{7.1}
\end{aligned}$$

- **One-hidden layer neural network model with fixed hidden-to-output layer:**

By expanding the expression we obtain

$$\mathbb{E}_{\mathbb{Q}_{\boldsymbol{\theta}_T}}[\|\widehat{\boldsymbol{y}}_T - \boldsymbol{y}_T\|_{\ell_2}^2] = \mathbb{E}[\|\boldsymbol{V}\varphi(\widehat{\boldsymbol{W}}_T \boldsymbol{x}_T) - \boldsymbol{V}\varphi(\boldsymbol{W}_T \boldsymbol{x}_T)\|_{\ell_2}^2] + k\sigma^2$$

$$\geq \sigma_{\min}^2(\boldsymbol{V})\mathbb{E}[\|\varphi(\widehat{\boldsymbol{W}}_T \boldsymbol{x}_T) - \varphi(\boldsymbol{W}_T \boldsymbol{x}_T)\|_{\ell_2}^2] + k\sigma^2 \qquad (7.2)$$

Let $\boldsymbol{A} = \widehat{\boldsymbol{W}}_T \Sigma_T^{\frac{1}{2}}$, $\boldsymbol{B} = \boldsymbol{W}_T \Sigma_T^{\frac{1}{2}}$, and $\boldsymbol{x} = \Sigma_T^{\frac{-1}{2}} \boldsymbol{x}_T$. So $\boldsymbol{x} \sim \mathcal{N}(0, I_d)$. Moreover, let $\boldsymbol{A} = \begin{bmatrix} \boldsymbol{\alpha}_1^T \\ \vdots \\ \boldsymbol{\alpha}_\ell^T \end{bmatrix}$ and

$\boldsymbol{B} = \begin{bmatrix} \boldsymbol{\beta}_1^T \\ \vdots \\ \boldsymbol{\beta}_\ell^T \end{bmatrix}$. Since $\mathbb{E}[\|\varphi(\widehat{\boldsymbol{W}}_T \boldsymbol{x}_T) - \varphi(\boldsymbol{W}_T \boldsymbol{x}_T)\|_{\ell_2}^2] = \sum_{i=1}^\ell \mathbb{E}[|\varphi(\boldsymbol{\alpha}_i^T \boldsymbol{x}) - \varphi(\boldsymbol{\beta}_i^T \boldsymbol{x})|^2]$, it suffices
to find a lower bound for the following expression

$$\mathbb{E}[|\varphi(\boldsymbol{a}^T \boldsymbol{x}) - \varphi(\boldsymbol{b}^T \boldsymbol{x})|^2]$$

where $a$ and $b$ are two arbitrary vectors in $\mathbb{R}^d$, $\varphi$ is the ReLU activation function, and $\boldsymbol{x} \sim \mathcal{N}(0, I_d)$.
We have

$$\mathbb{E}[|\varphi(\boldsymbol{a}^T \boldsymbol{x}) - \varphi(\boldsymbol{b}^T \boldsymbol{x})|^2] = \mathbb{E}[|\varphi(\boldsymbol{a}^T \boldsymbol{x})|^2] + \mathbb{E}[|\varphi(\boldsymbol{b}^T \boldsymbol{x})|^2] - 2\mathbb{E}[\varphi(\boldsymbol{a}^T \boldsymbol{x})\varphi(\boldsymbol{b}^T \boldsymbol{x})]. \qquad (7.3)$$

Now we calculate each term appearing on the right hand side.

Since $\boldsymbol{a}^T \boldsymbol{x} \sim N(0, \|\boldsymbol{a}\|_{\ell_2}^2)$, we have

$$\mathbb{E}[|\varphi(\boldsymbol{a}^T \boldsymbol{x})|^2] = \mathbb{E}[|\mathrm{ReLU}(\boldsymbol{a}^T \boldsymbol{x})|^2]$$

$$= \int_0^{+\infty} \frac{t^2}{\sqrt{2\pi}\|\boldsymbol{a}\|_{\ell_2}} e^{\frac{-t^2}{2\|\boldsymbol{a}\|_{\ell_2}^2}} \, dt$$

$$= \frac{\|\boldsymbol{a}\|_{\ell_2}^2}{2}.$$

Similarly, $\mathbb{E}[|\varphi(\boldsymbol{b}^T \boldsymbol{x})|^2] = \frac{\|\boldsymbol{b}\|_{\ell_2}^2}{2}$. To calculate the cross term note that $\boldsymbol{a}^T \boldsymbol{x}$ and $\boldsymbol{b}^T \boldsymbol{x}$ are jointly
Gaussian with zero mean and covariance matrix equal to

$$\begin{bmatrix} \|\boldsymbol{a}\|_{\ell_2}^2 & \boldsymbol{a}^T \boldsymbol{b} \\ \boldsymbol{a}^T \boldsymbol{b} & \|\boldsymbol{b}\|_{\ell_2}^2 \end{bmatrix}.$$

Therefore, we have (e.g. see (18))

$$2\mathbb{E}[\varphi(\boldsymbol{a}^T \boldsymbol{x})\varphi(\boldsymbol{b}^T \boldsymbol{x})] = 2\mathbb{E}[\mathrm{ReLU}(\boldsymbol{a}^T \boldsymbol{x})\mathrm{ReLU}(\boldsymbol{b}^T \boldsymbol{x})]$$

$$= \|\boldsymbol{a}\|_{\ell_2} \|\boldsymbol{b}\|_{\ell_2} \frac{\sqrt{1-\gamma^2} + (\pi - \cos^{-1}(\gamma))\gamma}{\pi} \qquad (7.4)$$

where $\gamma := \frac{\boldsymbol{a}^T \boldsymbol{b}}{\|\boldsymbol{a}\|_{\ell_2}\|\boldsymbol{b}\|_{\ell_2}}$.
Plugging these results in (7.3), we can conlude that

$$\mathbb{E}[|\varphi(\boldsymbol{a}^T \boldsymbol{x}) - \varphi(\boldsymbol{b}^T \boldsymbol{x})|^2] = \frac{\|\boldsymbol{a}\|_{\ell_2}^2}{2} + \frac{\|\boldsymbol{b}\|_{\ell_2}^2}{2} - \|\boldsymbol{a}\|_{\ell_2}\|\boldsymbol{b}\|_{\ell_2} \frac{\sqrt{1-\gamma^2} + (\pi - \cos^{-1}(\gamma))\gamma}{\pi}$$

$$= \frac{1}{2}\|\boldsymbol{a} - \boldsymbol{b}\|_{\ell_2}^2 - \|\boldsymbol{a}\|_{\ell_2}\|\boldsymbol{b}\|_{\ell_2} \frac{\sqrt{1-\gamma^2} - \gamma\cos^{-1}(\gamma)}{\pi}. \qquad (7.5)$$

We are interested in finding a universal constant $0 < c < \frac{1}{2}$ such that $\mathbb{E}[|\varphi(\boldsymbol{a}^T\boldsymbol{x}) - \varphi(\boldsymbol{b}^T\boldsymbol{x})|^2] \geq c\,\|\boldsymbol{a} - \boldsymbol{b}\|_{\ell_2}^2$. Using (7.5) and dividing by $\|\boldsymbol{a}\|_{\ell_2}\|\boldsymbol{b}\|_{\ell_2}$ this is equivalent to finding $0 < c < \frac{1}{2}$ such that

$$\left(\frac{1}{2} - c\right)\frac{\|\boldsymbol{a}\|_{\ell_2}^2 + \|\boldsymbol{b}\|_{\ell_2}^2 - 2\boldsymbol{a}^T\boldsymbol{b}}{\|\boldsymbol{a}\|_{\ell_2}\|\boldsymbol{b}\|_{\ell_2}} + \frac{\gamma\cos^{-1}(\gamma) - \sqrt{1-\gamma^2}}{\pi} \geq 0$$

Next note that by the AM-GM inequality we have

$$\left(\frac{1}{2} - c\right)\frac{\|\boldsymbol{a}\|_{\ell_2}^2 + \|\boldsymbol{b}\|_{\ell_2}^2 - 2\boldsymbol{a}^T\boldsymbol{b}}{\|\boldsymbol{a}\|_{\ell_2}\|\boldsymbol{b}\|_{\ell_2}} + \frac{\gamma\cos^{-1}(\gamma) - \sqrt{1-\gamma^2}}{\pi}$$

$$\geq \left(\frac{1}{2} - c\right)\frac{2\|\boldsymbol{a}\|_{\ell_2}\|\boldsymbol{b}\|_{\ell_2} - 2\boldsymbol{a}^T\boldsymbol{b}}{\|\boldsymbol{a}\|_{\ell_2}\|\boldsymbol{b}\|_{\ell_2}} + \frac{\gamma\cos^{-1}(\gamma) - \sqrt{1-\gamma^2}}{\pi}$$

$$= \left(\frac{1}{2} - c\right)(2 - 2\gamma) + \frac{\gamma\cos^{-1}(\gamma) - \sqrt{1-\gamma^2}}{\pi}$$

$$= (1 - \gamma)\left[(1 - 2c) + \frac{1}{\pi}\cdot\frac{\gamma\cos^{-1}(\gamma) - \sqrt{1-\gamma^2}}{1-\gamma}\right].$$

Therefore, it suffices to find $0 < c < \frac{1}{2}$ such that the R.H.S. of the above is positive. It is easy to verify that $h(\gamma) := \frac{\gamma\cos^{-1}(\gamma) - \sqrt{1-\gamma^2}}{1-\gamma} \geq \frac{-\pi}{2}$ for $-1 \leq \gamma < 1$. This in turn implies that the R.H.S. above is positive with $c = \frac{1}{4}$.

In the case when $\|\boldsymbol{a}\|_{\ell_2} = 0$ or $\|\boldsymbol{b}\|_{\ell_2} = 0$ ( let us assume $\|\boldsymbol{b}\|_{\ell_2} = 0$), (7.3) reduces to

$$\mathbb{E}[|\varphi(\boldsymbol{a}^T\boldsymbol{x}) - \varphi(\boldsymbol{b}^T\boldsymbol{x})|^2] = \mathbb{E}[|\varphi(\boldsymbol{a}^T\boldsymbol{x})|^2]$$

$$= \frac{\|\boldsymbol{a}\|_{\ell_2}^2}{2}$$

$$\geq \frac{1}{2}\|\boldsymbol{a} - \boldsymbol{b}\|_{\ell_2}^2$$

$$\geq \frac{1}{4}\|\boldsymbol{a} - \boldsymbol{b}\|_{\ell_2}^2.$$

Plugging the latter into (7.2) we arrive at

$$\mathbb{E}_{\mathbb{Q}_{\boldsymbol{\theta}_T}}[\|\widehat{\boldsymbol{y}}_T - \boldsymbol{y}_T\|_{\ell_2}^2] \geq \sigma_{\min}^2(\boldsymbol{V})\mathbb{E}\left[\left\|\varphi(\widehat{\boldsymbol{W}}_T\boldsymbol{x}_T) - \varphi(\boldsymbol{W}_T\boldsymbol{x}_T)\right\|_{\ell_2}^2\right] + k\sigma^2$$

$$\geq \frac{1}{4}\sigma_{\min}^2(\boldsymbol{V})\|\boldsymbol{\Sigma}_T^{\frac{1}{2}}(\widehat{\boldsymbol{W}}_T - \boldsymbol{W}_T)^T\|_F^2 + k\sigma^2,$$

concluding the proof.

• **One-hidden layer neural network model with fixed input-to-hidden layer:**

By expanding the expression we get

$$\mathbb{E}_{\mathbb{Q}_{\boldsymbol{\theta}_T}}[\|\widehat{\boldsymbol{y}}_T - \boldsymbol{y}_T\|_{\ell_2}^2] = \mathbb{E}\left[\left\|\widehat{\boldsymbol{V}}_T\varphi(\boldsymbol{W}\boldsymbol{x}_T) - \boldsymbol{V}_T\varphi(\boldsymbol{W}\boldsymbol{x}_T)\right\|_{\ell_2}^2\right] + k\sigma^2.$$

If we denote $\mathbb{E}[\varphi(\boldsymbol{W}\boldsymbol{x}_T)\varphi(\boldsymbol{W}\boldsymbol{x}_T)^T] = \widetilde{\boldsymbol{\Sigma}}_T$, then similar to (7.1) we obtain

$$\mathbb{E}_{\mathbb{Q}_{\boldsymbol{\theta}_T}}[\|\widehat{\boldsymbol{y}}_T - \boldsymbol{y}_T\|_{\ell_2}^2] = \|\widetilde{\boldsymbol{\Sigma}}_T^{\frac{1}{2}}(\widehat{\boldsymbol{V}}_T - \boldsymbol{V}_T)^T\|_F^2 + k\sigma^2. \tag{7.6}$$

Therefore, it suffices to calculate $\widetilde{\boldsymbol{\Sigma}}_T$. Let $\boldsymbol{W}\boldsymbol{\Sigma}_T^{\frac{1}{2}} = \begin{bmatrix} \boldsymbol{a}_1^T \\ \vdots \\ \boldsymbol{a}_\ell^T \end{bmatrix}$ and $\boldsymbol{x} = \boldsymbol{\Sigma}_T^{\frac{-1}{2}}\boldsymbol{x}_T$ (so $\boldsymbol{x} \sim \mathcal{N}(0, I_d)$). By (7.4) we obtain that

$$\widetilde{\boldsymbol{\Sigma}}_T = \left[\frac{1}{2}\|\boldsymbol{a}_i\|_{\ell_2}\|\boldsymbol{a}_j\|_{\ell_2}\frac{\sqrt{1-\gamma_{ij}^2} + (\pi - \cos^{-1}(\gamma_{ij}))\gamma_{ij}}{\pi}\right]_{ij} \tag{7.7}$$

where $\gamma_{ij} := \frac{\boldsymbol{a}_i^T\boldsymbol{a}_j}{\|\boldsymbol{a}_i\|_{\ell_2}\|\boldsymbol{a}_j\|_{\ell_2}}$.

## 7.2 Calculating KL-Divergences for the Linear Model (Proof of Lemma 1)

First we compute the KL-divergence between the distributions $\mathbb{P}_{\boldsymbol{W}_S^{(i)}}(\boldsymbol{x}_S, \boldsymbol{y}_S)$ and $\mathbb{P}_{\boldsymbol{W}_S^{(j)}}(\boldsymbol{x}_S, \boldsymbol{y}_S)$:

$$
\begin{aligned}
D_{KL}(\mathbb{P}_{\boldsymbol{W}_S^{(i)}}(\boldsymbol{x}_S, \boldsymbol{y}_S), \mathbb{P}_{\boldsymbol{W}_S^{(j)}}(\boldsymbol{x}_S, \boldsymbol{y}_S)) =& D_{KL}(\mathbb{P}_{\boldsymbol{W}_S^{(i)}}(\boldsymbol{x}_S), \mathbb{P}_{\boldsymbol{W}_S^{(j)}}(\boldsymbol{x}_S)) \\
&+ \mathbb{E}[D_{KL}(\mathbb{P}_{\boldsymbol{W}_S^{(i)}}(\boldsymbol{y}_S|\boldsymbol{x}_S), \mathbb{P}_{\boldsymbol{W}_S^{(j)}}(\boldsymbol{y}_S|\boldsymbol{x}_S))].
\end{aligned}
$$

The marginal distributions $\mathbb{P}_{\boldsymbol{W}_S^{(i)}}(\boldsymbol{x}_S)$ and $\mathbb{P}_{\boldsymbol{W}_S^{(j)}}(\boldsymbol{x}_S)$ are equal so their KL-divergence is zero. The conditional distributions $\mathbb{P}_{\boldsymbol{W}_S^{(i)}}(\boldsymbol{y}_S|\boldsymbol{x}_S)$ and $\mathbb{P}_{\boldsymbol{W}_S^{(j)}}(\boldsymbol{y}_S|\boldsymbol{x}_S)$ are normally distributed with covariance matrix $\sigma^2 \mathbf{I}_k$ and with mean respectively equal to $\boldsymbol{W}_S^{(i)} \boldsymbol{x}_S$ and $\boldsymbol{W}_S^{(j)} \boldsymbol{x}$. Therefore,

$$
D_{KL}(\mathbb{P}_{\boldsymbol{W}_S^{(i)}}(\boldsymbol{y}_S|\boldsymbol{x}_S), \mathbb{P}_{\boldsymbol{W}_S^{(j)}}(\boldsymbol{y}_S|\boldsymbol{x}_S)) = \frac{\left\| \boldsymbol{W}_S^{(i)} \boldsymbol{x}_S - \boldsymbol{W}_S^{(j)} \boldsymbol{x}_S \right\|_{\ell_2}^2}{2\sigma^2}.
$$

This in turn implies that

$$
D_{KL}(\mathbb{P}_{\boldsymbol{W}_S^{(i)}}(\boldsymbol{x}_S, \boldsymbol{y}_S), \mathbb{P}_{\boldsymbol{W}_S^{(j)}}(\boldsymbol{x}_S, \boldsymbol{y}_S)) = \frac{\mathbb{E}[\left\| \boldsymbol{W}_S^{(i)} \boldsymbol{x}_S - \boldsymbol{W}_S^{(j)} \boldsymbol{x}_S \right\|_{\ell_2}^2]}{2\sigma^2} = \frac{\| \boldsymbol{\Sigma}_S^{\frac{1}{2}} (\boldsymbol{W}_S^{(i)} - \boldsymbol{W}_S^{(j)})^T \|_F^2}{2\sigma^2},
$$

where the last equality follows similarly to the proof of Proposition 1 in the linear case.

A similar calculation also yields

$$
D_{KL}(\mathbb{Q}_{\boldsymbol{W}_T^{(i)}}(\boldsymbol{x}_T, \boldsymbol{y}_T), \mathbb{Q}_{\boldsymbol{W}_T^{(j)}}(\boldsymbol{x}_T, \boldsymbol{y}_T)) = \frac{\| \boldsymbol{\Sigma}_T^{\frac{1}{2}} (\boldsymbol{W}_T^{(i)} - \boldsymbol{W}_T^{(j)})^T \|_F^2}{2\sigma^2}.
$$

## 7.3 Lower Bound for Minimax Risk When $\Delta \geq \sqrt{\frac{\sigma^2 D \log 2}{r_T n_T}}$ and $\Delta < \sqrt{\frac{\sigma^2 D \log 2}{r_T n_T}}$ ( Proof of Lemma 2)

Consider the set

$$
\left\{ \eta : \eta = \boldsymbol{\Sigma}_T^{\frac{1}{2}} \boldsymbol{W}_T^T \text{ for some } \mathbf{W}_T \in \mathbb{R}^{k \times d} \text{ and } \|\eta\|_F \leq 4\delta \right\}
$$

where $\delta > 0$ is a value to be determined later in the proof. Furthermore, let $\{\eta^1, ..., \eta^N\}$ be a $2\delta$-packing of this set in the $F$-norm. Since $\dim(\text{range}(\boldsymbol{\Sigma}_T^{\frac{1}{2}} \boldsymbol{W}_T^T)) = rk$ in which $\boldsymbol{W}_T$ is regarded as an input, this set sits in a space of dimension $rk$, where $r = \text{rank}(\boldsymbol{\Sigma}_T)$. Hence we can find such a packing with $\log N \geq rk \log 2$ elements.

Therefore, we have a collection of matrices of the form $\eta^j = \boldsymbol{\Sigma}_T^{\frac{1}{2}} (\boldsymbol{W}_T^{(i)})^T$ for some $\boldsymbol{W}_T^{(i)} \in \mathbb{R}^{k \times d}$ such that

$$
\left\| \boldsymbol{\Sigma}_T^{\frac{1}{2}} (\boldsymbol{W}_T^{(i)})^T \right\|_F \leq 4\delta \text{ for each } i \in [N]
$$

and

$$
2\delta \leq \| \boldsymbol{\Sigma}_T^{\frac{1}{2}} (\boldsymbol{W}_T^{(i)} - \boldsymbol{W}_T^{(j)})^T \|_F \leq 8\delta \text{ for each } i \neq j \in [N] \times [N].
$$

So by Lemma 1 we get

$$
D_{KL}(\mathbb{Q}_{\boldsymbol{W}_T^{(i)}}, \mathbb{Q}_{\boldsymbol{W}_T^{(j)}}) \leq \frac{32\delta^2}{\sigma^2} \text{ for each } i \neq j \in [N] \times [N].
$$

Then set $\delta \leq \frac{\Delta}{8}$. We can choose $\mathbb{P}_{\boldsymbol{W}_S^{(1)}} = ... = \mathbb{P}_{\boldsymbol{W}_S^{(N)}}$ with $\boldsymbol{W}_S^{(1)} = ... = \boldsymbol{W}_S^{(N)}$ and $\|\boldsymbol{\Sigma}_T^{\frac{1}{2}} (\boldsymbol{W}_S^{(1)})^T \|_F = 4\delta$ . So they satisfy the condition

$$
\rho(\boldsymbol{W}_S^{(i)}, \boldsymbol{W}_T^{(i)}) \leq 8\delta \leq \Delta \text{ for each } i \in [N].
$$

Figure 6 illustrates this configuration. So having samples from $\mathbb{P}_{\boldsymbol{W}_S^{(i)}}$ does not contain any informa-

Figure 6: Configuration of the parameters of source and target distributions in Lemma 2.

tion about the true index in $[N]$ in the hypothesis testing problem and $E$ is independent of $J$, hence we get

$$I(J; E) = 0.$$

Therefore, using (6.2) and (6.3) we get

$$\mathcal{R}_T(\mathcal{P}_\Delta; \phi \circ \rho) \geq \phi(\delta) \left( 1 - \frac{n_T \frac{32\delta^2}{\sigma^2} + \log 2}{rk \log 2} \right)$$

$$= \delta^2 \left( 1 - \frac{n_T \frac{32\delta^2}{\sigma^2} + \log 2}{rk \log 2} \right)$$

for any $0 \leq \delta \leq \frac{\Delta}{8}$. We need to check the boundary and stationary points to solve the above optimization problem.

If $\Delta \geq \sqrt{\frac{\sigma^2 (rk-1) \log 2}{n_T}} = \sqrt{\frac{\sigma^2 D \log 2}{n_T}}$ holds, then

$$\mathcal{R}_T(\mathcal{P}_\Delta; \phi \circ \rho) \geq \frac{\sigma^2 (rk-1)^2 \log 2}{128 n_T rk}.$$

Since $D = rk - 1 \geq 20$, we have $\frac{\log 2}{rk} \geq \frac{1}{2(rk-1)}$, so

$$\mathcal{R}_T(\mathcal{P}_\Delta; \phi \circ \rho) \geq \frac{\sigma^2 D}{256 n_T} \tag{7.8}$$

and if $\Delta < \sqrt{\frac{\sigma^2 (rk-1) \log 2}{n_T}} = \sqrt{\frac{\sigma^2 D \log 2}{n_T}}$ then

$$\mathcal{R}_T(\mathcal{P}_\Delta; \phi \circ \rho) \geq \left( \frac{\Delta}{8} \right)^2 \left[ 1 - \frac{\frac{n_T \Delta^2}{2\sigma^2} + \log 2}{rk \log 2} \right]$$

$$\geq \left( \frac{\Delta}{8} \right)^2 \left[ 1 - \frac{\frac{n_T \Delta^2}{2\sigma^2} + \log 2}{D \log 2} \right].$$

Since $D \geq 20$ we get

$$\mathcal{R}_T(\mathcal{P}_\Delta; \phi \circ \rho) \geq \frac{1}{100} \Delta^2 \left[ 1 - 0.8 \frac{n_T \Delta^2}{\sigma^2 D} \right]. \tag{7.9}$$

## 7.4 Lower Bound for Minimax Risk When $\Delta \leq \frac{1}{45} \sqrt{\frac{\sigma^2 D}{r_S n_S + r_T n_T}}$ ( Proof of Lemma 3)

Let $\delta' = \Delta + \underbrace{u\Delta}_{=\delta}$, for $u > 0$ to be determined, Consider the set

$$\{ \eta : \eta = \Sigma_T^{\frac{1}{2}} W_S^T \text{ for some } \mathbf{W}_S \in \mathbb{R}^{k \times d} \text{ and } \|\eta\|_F \leq 4\delta' \}$$

and let $\{\eta^1, ..., \eta^N\}$ be a $2\delta'$-packing in the $F$-norm and consider each $\eta^i$ as a single point. Since $\dim(\text{range}(\Sigma_T^{\frac{1}{2}} W_T^T)) = rk$ in which $W_T$ is regarded as an input, this set sits in a space of dimension $rk$ where $r = \text{rank}(\Sigma_T)$. Therefore, we can find such a packing with $\log N \geq rk \log 2$ elements.

Hence, we have a collection of matrices of the form $\eta^i = \Sigma_T^{\frac{1}{2}} (W_S^{(i)})^T$ for some $W_S^{(i)} \in \mathbb{R}^{k \times d}$ such that

$$\|\Sigma_T^{\frac{1}{2}} (W_S^{(i)})^T\|_F \leq 4\delta' \text{ for each } i \in [N]$$

$$2\delta' \leq \|\Sigma_T^{\frac{1}{2}} (W_S^{(i)} - W_S^{(j)})^T\|_F \leq 8\delta' \text{ for each } i \neq j \in [N] \times [N].$$

Figure 7: Configuration of the parameters of source and target distributions in Lemma 3.

We choose each $W_T^{(i)}$ such that $\rho(W_T^{(i)}, M_S^{(i)}) = \|\Sigma_T^{\frac{1}{2}} (W_T^{(i)} - W_S^{(i)})^T\|_F = \Delta$. So

$$\rho(W_T^{(i)}, W_T^{(j)}) \geq 2\delta \text{ for each } i \neq j \in [N] \times [N].$$

Moreover,

$$\rho(W_T^{(i)}, W_T^{(j)}) \leq \rho(W_T^{(i)}, W_S^{(i)}) + \rho(W_S^{(i)}, M_S^{(j)}) + \rho(W_S^{(j)}, W_T^{(j)})$$
$$\leq 2\Delta + 8(\Delta + u\Delta).$$

Figure 7 illustrates this configuration. By Lemma 1 we have

$$D_{KL}(\mathbb{Q}_{W_T^{(i)}}, \mathbb{Q}_{W_T^{(j)}}) \leq \frac{2\Delta^2(5 + 4u)^2}{\sigma^2} \text{ for each } i \neq j \in [N] \times [N].$$

Also we have

$$\|\Sigma_S^{\frac{1}{2}} (W_S^{(i)} - W_S^{(j)})^T\|_F = \|\Sigma_S^{\frac{1}{2}} \Sigma_T^{-\frac{1}{2}} \Sigma_T^{\frac{1}{2}} (W_S^{(i)} - W_S^{(j)})^T\|_F$$
$$\leq 8 \left\| \Sigma_S^{\frac{1}{2}} \Sigma_T^{-\frac{1}{2}} \right\| \delta'$$
$$= 8 \left\| \Sigma_S^{\frac{1}{2}} \Sigma_T^{-\frac{1}{2}} \right\| (\Delta + u\Delta)$$

Hence, by Lemma 1 we have

$$D_{KL}(\mathbb{P}_{W_S^{(i)}}, \mathbb{P}_{W_S^{(j)}}) \leq \frac{32 \left\| \Sigma_S^{\frac{1}{2}} \Sigma_T^{-\frac{1}{2}} \right\|^2 \Delta^2 (u+1)^2}{\sigma^2} \text{ for each } i \neq j \in [N] \times [N].$$

Therefore, using (6.2) and (6.3) we arrive at

$$\mathcal{R}_T(\mathcal{P}; \phi \circ \rho) \geq \phi(u\Delta)\Big[1 - \frac{n_S \frac{32\left\|\Sigma_S^{\frac{1}{2}}\Sigma_T^{-\frac{1}{2}}\right\|^2 \Delta^2 (u+1)^2}{\sigma^2} + n_T \frac{2\Delta^2 (5+4u)^2}{\sigma^2} + \log 2}{rk \log 2}\Big]$$

$$= (u\Delta)^2 \Big[1 - \frac{n_S r_S \frac{32\Delta^2 (u+1)^2}{\sigma^2} + n_T r_T \frac{2\Delta^2 (5+4u)^2}{\sigma^2} + \log 2}{rk \log 2}\Big].$$

The above inequality holds for every $u \geq 0$. Maximizing the expression above over $u$ we can conclude if $\Delta \leq \sqrt{\frac{\sigma^2 (rk-1)\log 2}{32 n_S r_S + 50 n_T r_T}}$ then

$$\mathcal{R}_T(\mathcal{P}; \phi \circ \rho) \geq (u\Delta)^2 \Big[1 - \frac{n_S r_S \frac{32\Delta^2 (u+1)^2}{\sigma^2} + n_T r_T \frac{2\Delta^2 (5+4u)^2}{\sigma^2} + \log 2}{rk \log 2}\Big]$$

where $u = \frac{3\Delta(4n_S r_S + n_T r_T) + \sqrt{\Delta^2 [16(n_S r_S)^2 + 25(n_T r_T)^2 + 32 n_S r_S n_T r_T] + 4(n_S r_S + n_T r_T)(rk-1)\sigma^2 \log 2}}{16\Delta(n_\mathbb{P} r_\mathbb{P} + n_\mathbb{Q})}$.

Now, we need to simplify the above expressions. First note that

$$(u\Delta) \geq \Big(\frac{3\Delta}{16} + \frac{\sqrt{\Delta^2 + 4\frac{D\sigma^2 \log 2}{n_S r_S + n_T r_T}}}{16}\Big),$$

so

$$(u\Delta)^2 \geq \frac{\Delta^2 + 2.7\frac{D\sigma^2}{n_S r_S + n_T r_T}}{256}. \tag{7.10}$$

Moreover,

$$1 - \frac{n_S r_S \frac{32\Delta^2 (u+1)^2}{\sigma^2} + n_T \frac{2\Delta^2 (5+4u)^2}{\sigma^2} + \log 2}{rk \log 2} \geq 1 - \frac{[n_S r_S + n_T r_T]\frac{32\Delta^2 (\frac{5}{4}+u)^2}{\sigma^2} + \log 2}{D \log 2}$$

and

$$\Delta\Big(\frac{5}{4} + u\Big) \leq 2\Delta + \frac{1}{16}\sqrt{25\Delta^2 + \frac{4\log 2 D\sigma^2}{n_S r_S + n_T r_T}}.$$

Since $\Delta \leq \frac{1}{45}\sqrt{\frac{\sigma^2 D}{n_S r_S + n_T r_T}}$,

$$\Delta^2\Big(\frac{5}{4} + u\Big)^2 \leq \Big(4 + \Big(\frac{5}{16}\Big)^2\Big)\Delta^2 + \frac{1}{4}\Delta\sqrt{25\Delta^2 + \frac{4\log 2 D\sigma^2}{n_S r_S + n_T r_T}}$$

$$\leq \Big(4 + \Big(\frac{5}{16}\Big)^2\Big)\frac{1}{45^2}\frac{D\sigma^2}{n_S r_S + n_T r_T} + \frac{1}{4 \times 45^2}\sqrt{25^2 + 45^2 \times 4\log 2}\frac{D\sigma^2}{n_S r_S + n_T r_T}$$

$$\leq 0.012\frac{D\sigma^2}{n_S r_S + n_T r_T}.$$

Hence,

$$1 - \frac{[n_S r_S + n_T r_T]\frac{32\Delta^2 (\frac{5}{4}+u)^2}{\sigma^2} + \log 2}{D \log 2} \geq 1 - 0.56 - \frac{1}{D}$$

$$\geq 0.39.$$

Therefore, we arrive at

$$\mathcal{R}_T(\mathcal{P}; \phi \circ \rho) \geq \frac{\Delta^2}{1000} + \frac{6}{1000}\frac{D\sigma^2}{n_S r_S + n_T r_T}. \tag{7.11}$$

## 7.5 Proof of Theorem 1 (One-hidden layer neural network with fixed hidden-to-output layer)

By Proposition 1, the generalization error is bounded from below as

$$\mathbb{E}_{\mathbb{Q}_{\boldsymbol{\theta}_T}}\big[\|\widehat{\boldsymbol{y}}_T - \boldsymbol{y}_T\|_{\ell_2}^2\big] \geq \frac{1}{4}\sigma_{\min}^2(\boldsymbol{V})\|\boldsymbol{\Sigma}_T^{\frac{1}{2}}(\widehat{\boldsymbol{W}}_T - \boldsymbol{W}_T)^T\|_F^2 + k\sigma^2.$$

Therefore, it suffices to find a lower bound for the following quantity:

$$\mathcal{R}_T(\mathcal{P}_\Delta; \phi \circ \rho)$$
$$:= \inf_{\widehat{\boldsymbol{W}}_T} \sup_{(\mathbb{P}_{\boldsymbol{W}_S}, \mathbb{Q}_{\boldsymbol{W}_T}) \in \mathcal{P}_\Delta} \mathbb{E}_{S_{\mathbb{P}_{\boldsymbol{W}_S}} \sim \mathbb{P}_{\boldsymbol{W}_S}^{1:n_\mathbb{P}}} \big[\mathbb{E}_{S_{\mathbb{Q}_{\boldsymbol{W}_T}} \sim \mathbb{Q}_{\boldsymbol{W}_T}^{1:n_\mathbb{Q}}} \big[\phi(\rho(\widehat{\boldsymbol{W}}_T(S_{\mathbb{P}_{\boldsymbol{W}_S}}, S_{\mathbb{Q}_{\boldsymbol{W}_T}}), \boldsymbol{W}_T))\big]\big]$$

where $\phi(x) = x^2$ for $x \in \mathbb{R}$ and $\rho$ is defined per Definition 1. The rest of the proof is similar to the linear case as the corresponding transfer distance metrics are the same. We only need to upper bound the corresponding KL-divergences in this case. We do so by the following lemma.

**Lemma 4** *Suppose that $\mathbb{P}_{\boldsymbol{W}_S^{(i)}}$ and $\mathbb{P}_{\boldsymbol{W}_S^{(j)}}$ are the joint distributions of features and labels in a source task and $\mathbb{Q}_{\boldsymbol{W}_T^{(i)}}$ and $\mathbb{Q}_{\boldsymbol{W}_T^{(j)}}$ are joint distributions of features and labels in a target task as defined in Section 2.1 in the one-hidden layer neural network with fixed hidden-to-output layer model. Then $D_{KL}(\mathbb{P}_{\boldsymbol{W}_S^{(i)}} \| \mathbb{P}_{\boldsymbol{W}_S^{(j)}}) \leq \frac{\|\boldsymbol{V}\|^2 \|\boldsymbol{\Sigma}_S^{\frac{1}{2}}(\boldsymbol{W}_S^{(i)} - \boldsymbol{W}_S^{(j)})^T\|_F^2}{2\sigma^2}$ and $D_{KL}(\mathbb{Q}_{\boldsymbol{W}_T^{(i)}} \| \mathbb{Q}_{\boldsymbol{W}_T^{(j)}}) \leq \frac{\|\boldsymbol{V}\|^2 \|\boldsymbol{\Sigma}_S^{\frac{1}{2}}(\boldsymbol{W}_T^{(i)} - \boldsymbol{W}_T^{(j)})^T\|_F^2}{2\sigma^2}$.*

Furthermore, we also note that since in this case $\boldsymbol{W}_S, \boldsymbol{W}_T \in \mathbb{R}^{\ell \times d}$, the definition of $D$ is slightly different from that in the linear case. In this case $D = \text{rank}(\boldsymbol{\Sigma}_T)\ell - 1$.

## 7.6 Bounding the KL-Divergences in the Neural Network Model (Proof of Lemma 4)

First we compute the KL-divergence between the distributions $\mathbb{P}_{\boldsymbol{W}_S^{(i)}}(\boldsymbol{x}_S, \boldsymbol{y}_S)$ and $\mathbb{P}_{\boldsymbol{W}_S^{(j)}}(\boldsymbol{x}_S, \boldsymbol{y}_S)$

$$D_{KL}(\mathbb{P}_{\boldsymbol{W}_S^{(i)}}(\boldsymbol{x}_S, \boldsymbol{y}_S), \mathbb{P}_{\boldsymbol{W}_S^{(j)}}(\boldsymbol{x}_S, \boldsymbol{y}_S)) = D_{KL}(\mathbb{P}_{\boldsymbol{W}_S^{(i)}}(\boldsymbol{x}_S), \mathbb{P}_{\boldsymbol{W}_S^{(j)}}(\boldsymbol{x}_S))$$
$$+ \mathbb{E}[D_{KL}(\mathbb{P}_{\boldsymbol{W}_S^{(i)}}(\boldsymbol{y}_S | \boldsymbol{x}_S), \mathbb{P}_{\boldsymbol{W}_S^{(j)}}(\boldsymbol{y}_S | \boldsymbol{x}_S))].$$

The marginal distributions $\mathbb{P}_{\boldsymbol{M}_S^{(i)}}(\boldsymbol{x}_S)$ and $\mathbb{P}_{\boldsymbol{M}_S^{(j)}}(\boldsymbol{x}_S)$ are equal so their KL-divergence is zero. The conditional distributions $\mathbb{P}_{\boldsymbol{M}_S^{(i)}}(\boldsymbol{y}_S | \boldsymbol{x}_S)$ and $\mathbb{P}_{\boldsymbol{M}_S^{(j)}}(\boldsymbol{y}_S | \boldsymbol{x}_S)$ are normally distributed with covariance matrix $\sigma^2 \boldsymbol{I}_k$ and with mean respectively equal to $\boldsymbol{V}\varphi(\boldsymbol{W}_S^{(i)} \boldsymbol{x}_S)$ and $\boldsymbol{V}\varphi(\boldsymbol{W}_S^{(j)} \boldsymbol{x}_S)$. Therefore, we obtain

$$D_{KL}(\mathbb{P}_{\boldsymbol{W}_S^{(i)}}(\boldsymbol{y}_S | \boldsymbol{x}_S), \mathbb{P}_{\boldsymbol{W}_S^{(j)}}(\boldsymbol{y}_S | \boldsymbol{x}_S)) = \frac{\left\|\boldsymbol{V}\varphi(\boldsymbol{W}_S^{(i)} \boldsymbol{x}_S) - \boldsymbol{V}\varphi(\boldsymbol{W}_S^{(j)} \boldsymbol{x}_S)\right\|_{\ell_2}^2}{2\sigma^2}.$$

Then we have

$$D_{KL}(\mathbb{P}_{\boldsymbol{W}_S^{(i)}}(\boldsymbol{x}_S, \boldsymbol{y}_S), \mathbb{P}_{\boldsymbol{W}_S^{(j)}}(\boldsymbol{x}_S, \boldsymbol{y}_S)) = \frac{\mathbb{E}\left\|\boldsymbol{V}\varphi(\boldsymbol{W}_S^{(i)} \boldsymbol{x}_S) - \boldsymbol{V}\varphi(\boldsymbol{W}_S^{(j)} \boldsymbol{x}_S)\right\|_{\ell_2}^2}{2\sigma^2}$$
$$\leq \frac{\|\boldsymbol{V}\|^2 \|\boldsymbol{\Sigma}_S^{\frac{1}{2}}(\boldsymbol{W}_S^{(i)} - \boldsymbol{W}_S^{(j)})^T\|_F^2}{2\sigma^2}.$$

Since ReLU is a Lipschitz function. Similarly we get

$$D_{KL}(\mathbb{Q}_{\boldsymbol{W}_T^{(i)}}(\boldsymbol{x}_T, \boldsymbol{y}_T), \mathbb{Q}_{\boldsymbol{W}_T^{(j)}}(\boldsymbol{x}_T, \boldsymbol{y}_T)) = \frac{\mathbb{E}\|\boldsymbol{V}\varphi(\boldsymbol{W}_T^{(i)} \boldsymbol{x}_T) - \boldsymbol{V}\varphi(\boldsymbol{W}_T^{(j)} \boldsymbol{x}_T)\|_F^2}{2\sigma^2}$$
$$\leq \frac{\|\boldsymbol{V}\|^2 \|\boldsymbol{\Sigma}_T^{\frac{1}{2}}(\boldsymbol{W}_T^{(i)} - \boldsymbol{W}_T^{(j)})^T\|_F^2}{2\sigma^2}.$$

## 7.7 Proof of Theorem 1 (One-hidden layer neural network model with fixed input-to-hidden layer)

By Proposition 1, the generalization error is $\mathbb{E}_{\mathbb{Q}_{\boldsymbol{\theta}_T}}[\|\widehat{\boldsymbol{y}}_T - \boldsymbol{y}_T\|_{\ell_2}^2] = \|\widetilde{\boldsymbol{\Sigma}}_T^{\frac{1}{2}}(\widehat{\boldsymbol{V}}_T - \boldsymbol{V}_T)^T\|_F^2 + k\sigma^2$ so it suffices to find a lower bound for the following quantity

$$\mathcal{R}_T(\mathcal{P}_\Delta; \phi \circ \rho) := \inf_{\widehat{\boldsymbol{V}}_T} \sup_{(\mathbb{P}_{\boldsymbol{V}_S}, \mathbb{Q}_{\boldsymbol{V}_T}) \in \mathcal{P}_\Delta} \mathbb{E}_{S_{\mathbb{P}_{\boldsymbol{V}_S}} \sim \mathbb{P}_{\boldsymbol{V}_S}^{1:n_\mathbb{P}}} \left[\mathbb{E}_{S_{\mathbb{Q}_{\boldsymbol{V}_T}} \sim \mathbb{Q}_{\boldsymbol{V}_T}^{1:n_\mathbb{Q}}} \left[\phi(\rho(\widehat{\boldsymbol{V}}_T(S_{\mathbb{P}_{\boldsymbol{V}_S}}, S_{\mathbb{Q}_{\boldsymbol{V}_T}}), \boldsymbol{V}_T))\right]\right]$$

Where $\phi(x) = x^2$ for $x \in \mathbb{R}$ and $\rho$ is defined per Definition 1. Inherently, this case is the same as the linear model except that the distribution of the features has changed which was calculated in (7.7). The rest of the proof is similar to the linear case with the difference that $\boldsymbol{\Sigma}_S, \boldsymbol{\Sigma}_T$ should be replaced by $\widetilde{\boldsymbol{\Sigma}}_S, \widetilde{\boldsymbol{\Sigma}}_T$.

# 8 Appendix B

In this section we apply our theorem on DomainNet clipart and DomainNet sketch datasets to find a lower bound for target generalization error.

**DomainNet dataset.** First we pass the images of DomainNet-Clipart and DomainNet-Sketch through a ResNet-101 network pretrained on ImageNet with the fully connected top classifier removed and use the extracted features instead of the actual images.

**Training.** We trained a one-hidden layer neural network with 2048 input neurons, 40000 hidden neurons, and 345 output neurons for DomainNet-Clipart and DomainNet-Sketch dataset. We tuned the number of hidden neurons in order to get the best performance on the test dataset. We used MSE loss with one-hot encoded labels for training the networks. the trained network on DomainNet-Clipart has 98.4% train accuracy and 65.7% test accuracy and and the one trained on DomainNet-Sketch has 98.9% train accuracy and 51.2% test accuracy. Then we used the trained weights to estimate/calculate the parameters appearing in our lower bound. The noise levels are calculated based on the average loss of the trained ground truth models on the test dataset (note that this average loss equals $k\sigma^2 = 345\sigma^2$). Moreover, we estimated/calculated $r_s, r_T$, and the transfer distance, i.e. $\Delta$ as reported in Table 2.

| $\Delta$ | $k\sigma^2$ | $r_S$ | $r_T$ | $\sigma_{\min}^2(\boldsymbol{V})$ |
|----------|-------------|-------|-------|-----------------------------------|
| 202.15 | .54 | 2.247 | 2.58 | .249 |

Table 2: Estimated parameters of Theorem 1 for DomainNet clipart and Sketch datasets.

**Lower bound.** Using the values of Table 2, we get that $\Delta \geq \sqrt{\frac{D\sigma^2 \log 2}{r_T n_T}}$. Theorem 1 gives a lower bound for the generalization error which is illustrated in Figure 8. In Figure 9 we plot an upper bound for this dataset. The upper bound is obtained by empirical risk minimzation simply over target samples. In Table 3 we provide some exact values of upper and lower bound.

| Number of target samples | Lower bound | Upper bound |
|--------------------------|-------------|-------------|
| 5000 | 0.5424 | 1.0076 |
| 8150 | 0.5415 | 0.8573 |
| 15500 | 0.5408 | 0.7518 |
| 22150 | 0.5405 | 0.7266 |

Table 3: Some examples of exact values of lower and upper bound.

Figure 8: Theoretical lower bound on target generalization error as a function of the number of target samples for DomainNet dataset.

Figure 9: Upper bound on target generalization error as a function of the number of target samples for DomainNet dataset. The upper bound is obtained by simply minimizing the empirical risk over target samples.