[Reviews · NeurIPS 2020]

Review 1

Summary and Contributions: This paper developed a statistical minimax framework to characterize the fundamental limits of transfer learning for regression with linear and one-hidden layer neural network models. Specifically, a lower-bound for the target generalization error achievable by any algorithm was derived as a function of the number of labeled source and target data as well as appropriate notions of similarity between the source and target tasks. Various experiments are conducted to validate the theoretical finding.

Strengths: I like the problem studied in this paper; it is of sufficient interests to the audience of NeurIPS. Minimax risk will tell us what the fundamental limit is, i.e., what we cannot achieve in the scope of transfer learning. In minimax risk analysis, the supremum of risk needs to be taken over a proper class of transfer problems, and to this aim the authors propose the so-called transfer distance between source and target tasks to characterize the difficulty between different set of transfer learning problems.

Weaknesses: 1. I think the transfer distance can be interpreted as a measure of transferability, and the transfer distance defined in the paper seems to suggest that transfer learning is possible only when W_S and W_T are close to each other under the \Sigma_T norm. I understand that this definition is motivated from the proposition 1, but it is not always the case how people apply transfer learning in practice. In over-parametrized neural networks, two very different weights could both generate good performance model, but some learned features mappings can still be transferred to various tasks. Thus, I believe the transfer distance defined here does not fully characterize the transferability people discussed in general. 2. It is strange to me that the lower bound provided in Theorem 1 is not continuous as a function of \delta. Since the lower bound is not just characterizing the rate of the convergence, I would like to see the phase transition behavior of the bound between different regimes, and discontinuity would suggest that the lower bound is not tight at these points. Let’s consider an example where we fix \Delta>0 and n_T, and let n_S goes to infinity, then for large enough n_S, the lower bound will always operate in the moderate regime, which is not a function of n_S. Then we could find some choice of \Delta and n_T (like, \Delta=sqrt(sigma^2 D log 2/r_T n_T)), so that the bound value in the large distance regime is smaller than that of the moderate regime, which really confuses me. 3. I am not fully convinced by the empirical results in the paper. I would say it is nice to have these experiments as supplementary, but the contribution of this paper is the proof of the lower bound (converse), showing the performance of some achievable algorithms is not the correct way to validate the lower bound. I would suggest the authors to provide more intuitions in proving the lower bound (like the worst-case scenario) instead of these experiments, so that it will help people to come up with the minimax optimal transfer learning algorithm that could match the lower bound. I think the author's response had addressed my questions well. Tend to accept the paper.

Correctness: I did not check the proof provided in the supplementary, but the technical details described is the paper looks reasonable to me. As for the empirical part, I am not fully convinced, details in above.

Clarity: Yes, the paper is well written and easy to follow.

Relation to Prior Work: Prior works are addressed well.

Reproducibility: Yes

Additional Feedback:


Review 2

Summary and Contributions: This is a very insteresting paper showing minimax lower bounds for transfer learning in linear regression settings. Lower bound instance distribution: The authors consider the prototypical transfer learning problem where there is a source task available for learning another target task. - For both tasks, the input features are assumed to be drawn from the Gaussian distribution with certain covariance matrices. The labels follow a linear model with Gaussian noise that have the same noise level. - The goal is to derive the minimax rate for predicting the label of a target task sample. Main result: The minimax rate involves the noise part (naturally) plus another term B that varies according to the transfer distance of the two tasks. - When the transfer distance is large, B only scales with the effective dimension and the sample size of the target task. - When the transfer distance is small, B scales with the effective dimension and the sample size of both the source and target task. - When the transfer distance is in between, B interpolates the source/target task information. Proof approach: The proof follows by applying well techniques that reduce the lower bound to a hypothesis testing problem, and then carefully calculating the minimax risk depending on the transfer distance. ==================================== I have read the author's response and I stay with my initial review.

Strengths: - The main result provides a fine-grained minimax rate that takes into account how large the transfer distance is.

Weaknesses: - It is not clear whether this minimax rate is the best achievable within a more general family of hard instances. - It is also not clear what is the best achievable upper bound. That being said, I think this paper is making an interesting step towards the above goal(s).

Correctness: The proofs seem correct to me. One question: - How do you construct the 2\delta separated sets in the proof of the main result?

Clarity: There is no reference to put the introduction in context. In particular, all references appear in the related work paragraphs. I would suggest the authors to add citations to put their motivating sentences into the context of transfer learning literature (there are many!).

Relation to Prior Work: This paper also provides a minimax lower bound for a related multi-task learning setting in sparse regression. Oracle Inequalities and Optimal Inference under Group Sparsity Karim Lounici, Massimiliano Pontil, Alexandre B. Tsybakov, Sara van de Geer https://arxiv.org/abs/1007.1771

Reproducibility: Yes

Additional Feedback: - L185-L204: While I understand the intuition you are trying to convey in this discussion, it might be a bit misleading since the result is only a lower bound. I think you'll need a (matching) upper bound in order to make a more convincing claim here. It would be helpful to clarify this discussion a bit more. - The one-hidden-layer setting, where you either fix the hidden layer or the output layer, are both very artificial. Is there any scenario where these learning models actually arise? In addition, both of these settings just look like corollaries of the linear setting. It might be more clear to just state them as extensions.


Review 3

Summary and Contributions: The authors provide minimax lower bounds for the difficulty of transfer learning for a particular model. These bounds depend on a distance between the source and target models and the number of samples drawn from the source and target models. Empirical studies are presented to validate the bounds.

Strengths: Soundness: Both the theoretical result and the empirical studies appear sound. Significance and novelty: This paper provides quantitative bounds on the quality of transfer learning in terms of a quantitative distance between source and target. Given that there aren't many theoretical guarantees of this kind the paper is novel. Relevance to NeurIPS: definitely relevant

Weaknesses: Soundness: No weakness that I can see. The work appears correct. Significance and novelty: The models explored, linear and one layer networks, are limited. But the method introduced in this paper has the potential to be applied to more complicated models. Relevance to NeurIPS: no weakness

Correctness: The claims appear to be correct.

Clarity: The paper is clearly written.

Relation to Prior Work: The authors clearly discuss prior work and where their works fits.

Reproducibility: Yes

Additional Feedback:


Review 4

Summary and Contributions: This paper derives a lower-bound of the target generalization error for transfer learning focusing on linear and one-hidden layer neural network models. The bound captures the impact of the noise in the labels as well as transfer distance between source and target tasks. It also reveals the relationship between the lower-bound and the number of labeled training data available from the source and target domain under different settings of transfer distance. The paper also provides two experiments on real datasets and synthetic simulations to demonstrate the theoretical findings.

Strengths: This paper gives a clear and organized framework to derive the lower-bound of the target generalization error for transfer learning and analyzes impacts on the low-bounds of various parameters in detail, such as: transfer distance and the number of the available training data from the source and target domain. The lower-bound proposed in this paper is not based on a covariate shift assumption which requires and the source and target tasks to have the same best classifier and could be applied to the shallow one-hidden neural network where the hidden-output layer is fixed or the input-to hidden layer is fixed, which indicates limited significance and novelty of the contribution. The paper is relevant to the NeurIPS.

Weaknesses: To obtain the final lower-bound, the authors firstly give various definitions of parameters which are occurred in the final expression. However, they don’t explain what the exact meanings are and how to define them, such as, “Effective dimension” in Definition 3 and “Transfer coefficient” in Definition 4. Furthermore, the “sigma_min” in Proposition 1 and “A” in Definition 4 are not illustrated. Although the finding in this paper could be applied in the shallow neural network, it requires more strict constraints. For example, the neural network should be one-hidden layer and linear. More strictly, the hidden-to-output layer should be fixed or the input-to-hidden layer should be fixed, which limits the probability of extending the complex neural networks. Furthermore, the conclusion of this paper is direct and intuitionistic in the qualitative aspect. While in the quantitative aspect, it is hard to apply in the realistic datasets because of the complexity of the lower-bound.

Correctness: The paper obtains the final lower-bound by complex derivations, which conforms the intuitive understanding. In the subsection of ImageNet Experiments, the results couldn’t show the impacts on transfer target risk error of transfer distance and noisy level. It is suggested that keeping the noisy level is the same when measuring the impacts on transfer target risk error of transfer distance and vice versa. Furthermore, it is pleased to give the final lower-bounds of target generalization error in this realistic dataset and compare it with the target generalization error.

Clarity: The paper is well written. However, there exists a small mistake. “Deriving” should be “derive” in the line 77.

Relation to Prior Work: The paper provides clear illustrations to discuss the difference of some previous contributions. However, it is suggested that rewriting the first sentences in “Related work” since they are less relative to the paper. Investigating comprehensive and latest references is recommended because most of the relevant references are published before 2010.

Reproducibility: Yes

Additional Feedback:

[Author Response · NeurIPS 2020]

We thank the referees for a thorough reading of the manuscript and helpful comments.

**Reviewer 1(Weaknesses):** Our main goal is to develop a lower-bound for the target generalization error achievable
by any algorithm. We agree that our model/distance may not capture all practical scenarios but as our simulations
demonstrate it does seem to correlate well with algorithmic performance. **(1) Re overparam.n:** Thanks for the
very interesting question. Overparameterization does not pose a fundamental problem when using our notion of
distance in practice. There can indeed be many different W that generate the same output on the training data due to
overparam. However, in practice the $W$ found by GD is one that generalizes well. And while all $W$ with the same
training output are not close, all $W$ that generalize well must be close. This is because for such a $W$ we must have
$c \cdot \sigma_{\min}^2(V)\|\Sigma_S(W - W_S)\|_F^2 \leq \mathbb{E}\left[\|V\phi(Wx) - V\phi(W_Sx)\|^2\right] < \delta$. So even though GD may not find $W_S$ exactly
due to overparameterization (since it typically find a generalizable model) it must be close to $W_S$. In fact one can make
this rigorous using recent generalization theory (e.g. arxiv 1901.08584 and 1906.05392). Will further elaborate. **(2) Re**
**not continuous:** Our more elaborate bound in the proof of the main theorem (Sec. 6.3 457-459) are indeed continuous
at these transition points. To make the result more interpretable we simplified the expressions/loosened the bounds
which is the source of discontinuity. We will further clarify. **(3) Re empirical results:** Thanks for the suggestion. We
will move experiments to supp. and instead add more proof insights/mention some simple scenarios where our bounds
are tight.

**Reviewer 2: Re Weaknesses:** Our goal is to understand the fundamental limits of what is possible, hence the focus
on a lower bound. A lower bound is of significant practical interest in a variety of applications (see DARPA LwLL
program TA2) as it can help predict how much transferability from source to target is possible prior to committing
extensive resources to train complex transfer learning algorithms. Unfortunately, there is no good way to test the
sharpness of lower bounds numerically. We do expect our bounds to be tight up to numerical constants as they resemble
non-transfer learning bounds that are known to be sharp. In fact, very recently colleagues have informed us that they
have developed algorithms that achieve our lower bound up to a fixed constant (under a more restrictive covariate
shift assumption). To alleviate the reviewer's concern, in addition to mentioning this result (not yet publicly available)
we will also provide some simple instances/scenarios that demonstrate the sharpness of our bounds. We also hope to
develop matching algorithms in the general case in our future work. **Re question in correctness:** This is based on
slightly modifying corollary 4.2.13 in "High dimensional probability" by R. Vershynin. We will clarify. **Re refs in**
**Clarity:** Thanks, we will add more citations in the introduction section of the paper as well as discuss their pros and
cons w.r.t. Our paper. **Re Relation to prior work:** Thanks for suggesting this paper we will cite/add a discussion
about it.**Re Additional feedback):** **(1) re tightness:** Thanks, will add a discussion regarding the upper bound for the
risk as well as the tightness of our result. **(2) re more complex models:** We view the models discussed in the paper as
a first step towards studying more complicated neural network models. We do think it already captures some realistic
phenomena as demonstrated in our numerical experiments. That said, we are working on generalizing our result to
the case that both the hidden layer and output layers can both vary. **(3) Re are corollaries:** In Thm. 1, the result for
the third model (input-to-hidden fixed), is a corollary of the linear case. However, this is not the case re second model
(hidden-to-output fixed). In order to derive a lower bound in this case we need to find a metric for the risk and simplify
the generalization error and a major part of the proof is devoted to this purpose. Will further clarify.

**Reviewer 3:** Thanks a lot for the positive feedback/assessment.

**Reviewer 4: Re weaknesses: (1) re definitions:** Lower bounds (unlike upper bounds for algorithms) do not always
have easily interpretable quantities. That said, we have made an attempt to break our lower bound down into interpretable
and intuitive terms. All the terms are well defined by precise mathematical expressions and we have named them
accordingly to give some intuition what they would capture in the lower bound. We are happy to add more explanations
for these terms in the final version for further clarification. As for the parameter $A$ it should be replaced by $V$. Sorry
for the typo. We caught this typo right after the submission and in fact have highlighted the correct version in the first
paragraph of the supplementary. **(2) re strict constraints:** Indeed, our goal is to develop a lower bound that can be
applied to more complex models (which is a very challenging problem) and our goal is to provide an important first
step in this paper towards this goal. We note that even in the non-transfer learning scenario the theoretical study of
more complex models remains elusive. We would also like to note that the lower bound is in fact not hard to apply to
real datasets. All the parameters of the lower bound can be experimentally calculated/estimated from real data and
one can apply the lower bound in the practical setting by having access to a large enough number of samples as done
in our numerical experiments. In fact, we plan to participate in a challenge for such lower bounds (DARPA LwLL).
**(Correctness):** We did not understood the reviewer's concern. Note that in three of the experiments the noise level is
around the same and therefore the difference can only be attributed to the transfer distance. In a real experiment it is not
possible to keep the noise level exactly the same. We will however add some synthetic experiments to demonstrate this
further and to alleviate the reviewer's concern. We will also plot the final lower-bounds of target generalization error in
our experiment and compare it to the final target generalization error. **(Clarity):** Thanks for finding the typo. We will
fix it. **(Relation to prior work):** We will add further discussions on other related papers and their pros and cons.

[Meta-Review · NeurIPS 2020]

This paper addresses the problem of inductive transfer with one-hidden-layer neural networks or linear models and proposes minimax lower bounds for these models. Three reviewers and AC agree that it is a well written paper which studies an important problem. The proposed fine-grained minimax rate for transfer learning is a nice contribution to this field. Although the setting is somewhat simple, this work is inspiring for studying inductive transfer with neural networks. There are still some minor concerns on the organization of the paper and the evaluation of the proposed lower bound, which should be fully addressed in the camera-ready version. One reviewer with a negative score pointed out that while the finding in this paper could be applied in the shallow neural network, it requires more strict constraints which limit the probability of extending the complex neural networks. The AC also agrees with this concern, but believes that this paper is worth acceptance as a first step towards the challenging theoretical understanding of inductive transfer. The authors are certainly encouraged to further explore the boundary of this research direction.